# NITP: Next Implicit Token Prediction for LLM Pre-training

Xiangdong Zhang [1 2]   Debing Zhang [2]   Shaofeng Zhang [3]
Xiaohan Qin [1 2]   Yu Cheng [4]   Junchi Yan [1]

## Abstract

Standard Next-Token Prediction (NTP) supervises language models solely through discrete labels in the output logit space. We argue that this sparse, one-hot supervision leaves the latent representation space under-constrained, allowing hidden states to drift into degenerate and anisotropic configurations that limit generalization. To address this issue, we propose **Next Implicit Token Prediction (NITP)**, which augments discrete prediction with dense, continuous supervision directly in the representation space. NITP requires the model to predict the implicit semantic content of the next token, using shallow-layer representations from the same model as stable self-supervised targets. Theoretically, we show that NITP regularizes the optimization landscape by mitigating under-constrained degrees of freedom and enforcing a compact, structured representation geometry. Empirically, across dense and MoE models ranging from 0.5B to 9B parameters, NITP consistently **improves downstream performance with negligible computational overhead**. Notably, on the 9B MoE model, NITP achieves a **5.7%** absolute improvement on MMLU-Pro, along with gains of **6.4%** on C3 and **4.3%** on CommonsenseQA, with ∼**2%** additional training FLOPs and no additional inference cost. Our implementation is available at https://github.com/aHapBean/NITP.

## 1. Introduction

Large language models (LLMs) have achieved remarkable success through large-scale pre-training with the Next-Token Prediction (NTP) (Liu et al., 2024b; Yang et al., 2025;

[1]School of AI, Shanghai Jiao Tong University [2]Dots Studio, Xiaohongshu Inc. [3]University of Science and Technology of China [4]The Chinese University of Hong Kong. Correspondence to: Junchi Yan <yanjunchi@sjtu.edu.cn>.

*Proceedings of the 43$^{rd}$ International Conference on Machine Learning*, Seoul, South Korea. PMLR 306, 2026. Copyright 2026 by the author(s).

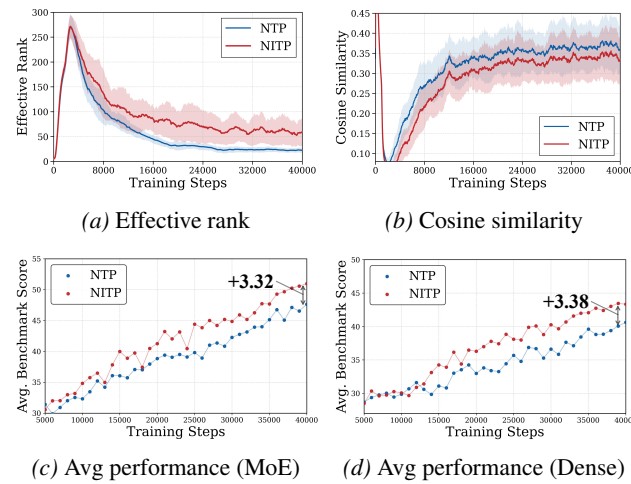

*(a)* Effective rank   *(b)* Cosine similarity

*(c)* Avg performance (MoE)   *(d)* Avg performance (Dense)

*Figure 1.* **Top:** Representation geometry of the last hidden states under NTP and NITP. **Bottom:** Average downstream performance of 9B MoE and 2B dense models (details in Appendix B).

Team et al., 2025; He & Su, 2025; Chen et al., 2024). By maximizing the likelihood of the next token over massive corpora, this paradigm learns general-purpose representations that support a wide range of downstream tasks (Guo et al., 2025). Although subsequent fine-tuning stages further adapt these models, their effectiveness critically depends on the representations learned during base pre-training (Yue et al., 2025). As a result, the pre-training objective plays a central role in determining the capability of LLMs.

Despite its empirical success, standard NTP provides supervision only through discrete one-hot targets in the output token space. While gradients propagate to hidden states via the output projection, the objective primarily constrains representations along the target logit direction, leaving many degrees of freedom in the latent space that remain weakly constrained. This raises a fundamental question: **Does next-token prediction alone sufficiently supervise the geometry of hidden representations?**

Prior work has identified a phenomenon known as *representation degeneration*, where likelihood-based training drives learned embeddings to collapse into a narrow, anisotropic cone (Ethayarajh, 2019; Wang et al., 2020; Barbero et al., 2024; Gao et al., 2019). Such geometric collapse limits the expressive capacity of representations and has been linked to

degraded generalization on downstream tasks (Ethayarajh, 2019; Zhao et al., 2024). To examine this issue, we track the geometric evolution of last hidden states throughout training. We utilize two established metrics: (1) Effective Rank (Roy & Vetterli, 2007), which quantifies the effective dimensionality of the utilized subspace, and (2) Average Cosine Similarity between paired tokens within training batches, serving as a proxy for the global anisotropy of the representation space. As illustrated in Figure 1, for NTP, the effective rank diminishes rapidly while cosine similarity increases, indicating that hidden representations drift toward degenerate, anisotropic configurations (detailed analysis in Appendix C). This confirms that standard NTP is geometrically under-constrained: it permits representations to sacrifice expressiveness and distinctiveness without penalty, as long as the next token is correctly predicted.

We attribute this behavior to the under-constrained nature of hidden representations under NTP, where the lack of geometric supervision allows representations to drift into low-expressivity configurations that may harm performance (Ethayarajh, 2019). To bridge this gap, we propose **Next Implicit Token Prediction** (NITP), an auxiliary pretraining objective that augments discrete next-token prediction with continuous supervision in the representation space. Instead of predicting only the token identity, NITP tasks the model with predicting an implicit token, defined as a latent semantic representation of the next token. We construct this supervision signal from the model's own shallow layers, which empirically preserve richer lexical and local semantic content than deeper, task-specialized layers (Liu et al., 2024c; Skean et al., 2025). This design enables scalable, self-supervised targets without introducing external encoders or additional annotations. As evidenced by Figure 1, NITP effectively mitigates representation degeneration, maintaining higher effective rank and preventing excessive anisotropy compared to standard NTP. **Our main contributions are summarized as follows:**

**1)** We analyze the training objective of NTP and show it provides **weak supervision for hidden representations**, allowing them to drift toward degenerate space.

**2)** We propose **Next Implicit Token Prediction** (NITP), a novel objective that augments next-token prediction with dense, continuous supervision in the latent space.

**3)** We introduce a **self-supervised design** for NITP by using shallow-layer representations as implicit tokens to serve as prediction targets, enabling semantically rich supervision.

**4)** We provide **theoretical analysis and empirical evidence** showing that NITP improves representation geometry and consistently enhances downstream performance.

## 2. Related Work

**Pre-training objectives for large language models.** The next-token prediction (NTP) objective is the dominant pretraining paradigm for modern large language models (Liu et al., 2024b; Yang et al., 2025), and has been shown to scale effectively with data and model size (Mann et al., 2020; Hoffmann et al., 2022; Kaplan et al., 2020). By optimizing the likelihood of the next discrete token given its context, NTP enables models to learn general-purpose representations that support a wide range of downstream tasks. To extend the predictive horizon beyond a single token, several works have explored multi-token prediction (MTP) (Gloeckle et al., 2024; Samragh et al., 2025; Liu et al., 2024b) and related objectives (Mahajan et al., 2025; Tack et al., 2025; Zuhri et al., 2025; Liu et al., 2025). These methods introduce auxiliary output heads to predict multiple future tokens or compressed summaries of future context, with the goal of improving planning or long-horizon modeling (Mahajan et al., 2025). Despite their differences, these approaches operate primarily in the discrete token space and rely on token-level supervision derived from one-hot targets. In contrast, our proposed Next Implicit Token Prediction departs from prior works by augmenting standard NTP with dense, continuous supervision at the representation level.

**Representation-level supervision.** Beyond discrete token-level objectives, prior work has explored supervising models directly in continuous representation spaces (Bengio et al., 2013; Oord et al., 2018; Jiao et al., 2020). A prominent direction is layer-wise distillation, which aligns intermediate representations between models to transfer structural or task-relevant knowledge (Romero et al., 2014; Sun et al., 2019; Jiao et al., 2020; Liang et al., 2023). This paradigm has recently been extended to generative LLMs (Sun et al., 2020b; Gu et al., 2024a; Xia et al., 2023). Notably, Sheared LLaMA (Xia et al., 2023) aligns hidden states to accelerate knowledge recovery after structured pruning. Another line of research focuses on self-supervised consistency, encouraging agreement between latent representations across time or views without discrete labels, *e.g.*, Contrastive Predictive Coding and BYOL (Oord et al., 2018; Grill et al., 2020), as well as self-distillation methods (Zhang et al., 2019; Lee et al., 2022). NITP differs from these approaches in both motivation and formulation: motivated by the under-constrained hidden states during LLM pre-training, NITP adopts an autoregressive objective in the latent space that requires the model to predict the semantic representation of the next token.

## 3. Methodology

In this section, we present **Next Implicit Token Prediction (NITP)**, a representation-level objective that explicitly supervises hidden states to complement standard Next-Token

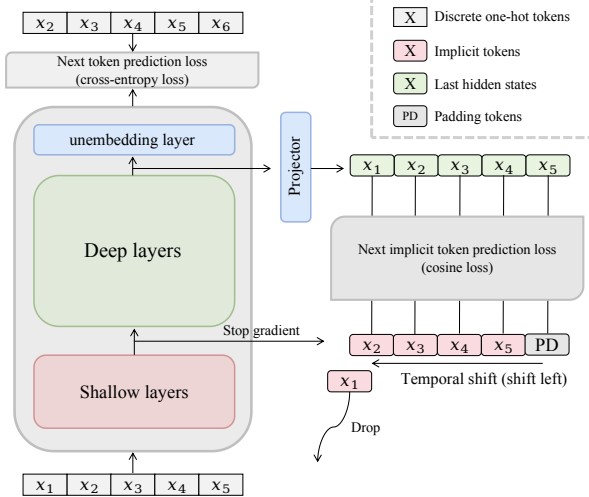

*Figure 2.* Overview of NITP. Next Implicit Token Prediction supervises hidden states by predicting temporally shifted implicit tokens, *i.e.* shallow-layer representations, and is jointly optimized with the standard next-token prediction objective. These shallow representations act as semantics anchors (see Section 3.3). The last-layer hidden states are projected to match the implicit targets using a cosine similarity loss.

Prediction (NTP), as illustrated in Figure 2. By encouraging the model to anticipate the semantic content of the next token in latent space, NITP regularizes representation geometry beyond discrete token-level supervision.

We first revisit the standard NTP paradigm and its limitations in supervising latent representations. We then introduce NITP, which trains the last hidden state $h_t$ to predict an implicit token—the shallow-layer representation of the $(t+1)$-th token with stop-gradient—using a cosine similarity loss jointly optimized with the NTP objective. As the implicit targets are self-generated during the forward pass, NITP incurs negligible computational overhead and scales naturally with model size (see Section 4 for results).

### 3.1. Preliminary: Next-Token Prediction

Given a token sequence $x = \{x_1, \ldots, x_T\}$, let $h_t \in \mathbb{R}^d$ denote the final hidden state at step $t$, serving as the contextual representation. Given an unembedding matrix $W \in \mathbb{R}^{|\mathcal{V}| \times d}$, the standard next-token prediction (NTP) objective minimizes the negative log-likelihood of the target token $x_{t+1}$:

$$\mathcal{L}_{\text{NTP}} = -\mathbb{E}_x \left[ \log \frac{\exp(h_t^\top w_{x_{t+1}})}{\sum_{j \in \mathcal{V}} \exp(h_t^\top w_j)} \right], \quad (1)$$

where $w_j$ denotes the embedding vector for token $j$. We omit layer normalization terms for brevity. While the NTP loss theoretically involves all vocabulary items via the normalization term, the gradient signal is empirically dominated by the target token $x_{t+1}$ and a few high-probability

contenders. Thus, the objective effectively constrains $h_t$ primarily along the direction of the target embedding $w_{x_{t+1}}$.

### 3.2. The Geometric Blind Spot of Next-Token Prediction

The gradient dominance described above creates a structural "blind spot" in the optimization landscape: the loss becomes near-invariant to perturbations within the vast subspace orthogonal to the target direction. Theoretically, this implies that a wide range of geometrically distinct latent states can yield identical token-level likelihoods.

In practice, this structural freedom does not yield diverse features but instead leads to *representation degeneration* (Ethayarajh, 2019; Wang et al., 2020; Barbero et al., 2024), where hidden states drift into a low-dimensional, anisotropic cone without penalty. We empirically confirm this collapse by tracking the geometric evolution of hidden states in Figure 1. Under standard NTP, we observe a rapid deterioration in Effective Rank (Roy & Vetterli, 2007) coinciding with a sharp rise in global Cosine Similarity. These trends indicate that the model sacrifices semantic richness for discriminative efficiency, compressing representations even as token-level accuracy improves.

From the perspective of the NTP objective, these degenerate configurations are valid as long as they preserve the correct logit ranking, yet they severely limit the expressive capacity of the latent space. Crucially, such a drop in representation expressiveness has been linked to degraded generalization on downstream tasks (Ethayarajh, 2019; Zhao et al., 2024). This observation underscores a critical limitation: **NTP defines *what* to predict, but fails to supervise *how* predictions are represented**, necessitating explicit guidance to enforce a semantically structured geometry. To this end, we propose Next Implicit Token Prediction (NITP), complementing NTP by explicitly supervising the latent representation space. Instead of constraining only discrete token identities, NITP requires the model to predict the implicit semantic representation of the next token, thereby directly supervising the geometry and evolution of hidden states.

### 3.3. Shallow Layers as Intrinsic Semantic Anchors

To construct an effective supervision target for NITP, we first clarify the semantic role of the last hidden state $h_t$. Prior works (Pal et al., 2023; Liu et al., 2024c) suggest that $h_t$ contains recoverable semantic information about the next token. Building on this view, we consider an idealized regime that the model predicts the correct next token $x_{t+1}$ with high confidence. In this setting, $h_t$ serves as the intermediate representation through which the prediction is realized, and thus is expected to encode semantic information relevant to $x_{t+1}$ in a form that supports the prediction task.

This observation naturally raises the question: what con-

stitutes an appropriate supervision target for $h_t$? A naive choice would be the static embedding of $x_{t+1}$. However, static embeddings suffer from *polysemy*—the same token (e.g., "bank") can correspond to vastly different meanings depending on the context. To account for this ambiguity, a suitable supervision target should be a *contextualized* representation that captures the precise semantics of the token conditioned on the context $x_{<t+1}$.

While external encoders (Raffel et al., 2020; Zhang et al., 2025) could provide such contextualized targets, they introduce prohibitive computational overhead and domain shifts. Instead, we propose an efficient, self-supervised solution: utilizing the model's own shallow layers as intrinsic semantic anchors. Our rationale is threefold:

- **Semantic richness:** Recent analysis (Skean et al., 2025) reveals that shallow layers act as powerful contextual encoders, resolving lexical ambiguity while preserving fine-grained semantic details. Studies (Liu et al., 2024c) explicitly demonstrate that semantic richness peaks in these early layers and degrades in deeper layers, as the model progressively discards details to focus solely on the sparse discriminative features required for NTP.

- **Stability:** Transformer training typically exhibits a bottom-up convergence pattern (Skean et al., 2025), where layers closer to the input stabilize much faster than deeper layers. This ensures that the generated implicit tokens are stable relative to the deep predictive state $h_t$.

- **Efficiency:** Utilizing shallow layers incurs negligible computational overhead. Unlike external models, the targets are extracted directly from intermediate activations computed during the standard forward pass.

By treating the semantic-rich, contextualized shallow representation of $x_{t+1}$ as the next implicit token, we impose a fidelity constraint that forces the deep state $h_t$ to maintain a structured geometry capable of containing full semantic details, thereby preventing representation degeneration.

### 3.4. Next Implicit Token Prediction

We introduce **Next Implicit Token Prediction (NITP)**, a continuous representation-level supervision objective designed to complement the discrete next-token prediction. While NTP predicts the discrete identity of the next token, NITP predicts its semantic essence in the latent space.

We formally distinguish between the predictive state $h_t$ and the prediction target $z_{t+1}$. We define the **implicit token** $z_{t+1}$ as the dense, contextualized semantic representation of the next token, derived from the model's own shallow layers. The NITP objective forces the final hidden state to align with this implicit token, thereby enforcing a structured and semantic representation geometry.

**Implicit token construction.** For a given input sequence $x$, let $E_{\text{shallow}}(\cdot)$ denote the forward pass through the first $k$ layers of the model (see Section 4.3 for details on the choice of $k$). We construct the implicit token $z_{t+1}$, which serves as the optimization target, by extracting the hidden representation at the future time step $t+1$:

$$z_{t+1} = \mathbf{sg}\left[E_{\text{shallow}}(x_{\leq t+1})^{(t+1)}\right] \in \mathbb{R}^d, \qquad (2)$$

where $(\cdot)^{(t+1)}$ denotes selecting the vector at position $t+1$, and $\mathbf{sg}[\cdot]$ indicates the stop-gradient operator. Crucially, by sourcing $z_{t+1}$ from shallow layers, we capture the contextualized semantics of the next token while avoiding the representation collapse often observed in deeper layers, making it a stable and semantically rich anchor.

**The prediction objective.** NITP tasks the last hidden state $h_t$ with predicting the implicit token $z_{t+1}$. To bridge the potential distributional gap between the deep predictive state and the shallow target, we employ a projection head $\mathcal{P}(\cdot)$ (*e.g.*, an MLP). The objective minimizes the geometric misalignment using cosine similarity:

$$\mathcal{L}_{\text{NITP}}(\theta) = 1 - \frac{\mathcal{P}(h_t)^\top z_{t+1}}{\|\mathcal{P}(h_t)\|_2 \cdot \|z_{t+1}\|_2}. \qquad (3)$$

We favor cosine similarity over other loss functions to enforce scale invariance as demonstrated in Section 4.3.

**Combined training.** The overall pre-training framework acts as a dual-supervision mechanism, combining the discrete NTP loss with the continuous NITP supervision:

$$\mathcal{L}_{\text{total}}(\theta) = \mathcal{L}_{\text{NTP}}(\theta) + \lambda\,\mathcal{L}_{\text{NITP}}(\theta), \qquad (4)$$

where $\lambda > 0$ is a hyperparameter controlling the weight of the implicit prediction. By requiring the model to predict both the discrete identity and the continuous implicit token, NITP explicitly regularizes the hidden representations toward a more semantically structured feature space.

### 3.5. Theoretical Analysis: Regularizing the Semantic Manifold

Standard Next-Token Prediction (NTP) supervises the hidden state $h_t$ primarily by maximizing its dot product with the target token embedding $w_{x_{t+1}}$. However, this objective is invariant to perturbations in the subspace orthogonal to the informative directions of the unembedding matrix. Mathematically, this manifests as a rank-deficient Hessian matrix with respect to the semantic geometry, allowing hidden states to drift into degenerate configurations without penalty—a phenomenon closely linked to representation anisotropy (Ethayarajh, 2019).

We formally analyze how Next Implicit Token Prediction (NITP) mitigates this by regularizing the angular geometry

of the latent space. Unlike NTP, which allows unconstrained drift in null directions, NITP acts as a geometric anchor, introducing positive curvature to the entire semantic subspace.

**Setup.** Let $h \in \mathbb{R}^d$ be the last hidden state used for implicit prediction. In practice, the NITP objective utilizes a non-linear projection $\mathcal{P}(\cdot)$. To characterize the regularization effect, we adopt the Generalized Gauss-Newton (GGN) approximation (Martens, 2020), which models the Hessian with respect to $h$ as $J_P^\top H_{\mathrm{NITP}} J_P$, where $H_{\mathrm{NITP}}$ denotes the Hessian of the NITP loss. Crucially, this quadratic form preserves the spectral properties (specifically positive definiteness) of $H_{\mathrm{NITP}}$, provided that the projector is locally well-conditioned (i.e., $J_P$ is full-rank). This implies that the regularization is fundamentally governed by the structure of the loss function. Therefore, we simplify the analysis by examining the NITP objective directly on $h$ to isolate its intrinsic geometric properties from the architectural details.

We decompose variations in $h$ into a *radial component* (change in norm) and an *angular component* (change in direction). Let $r = \|h\|_2$ and $u = h/r$. Let $z \in \mathbb{R}^d$ be the fixed implicit target derived from shallow layers, with normalized direction $v = z/\|z\|_2$. The NITP objective is defined as $\mathcal{L}_{\mathrm{NITP}}(h) = 1 - \cos(h, z) = 1 - u^\top v$.

**Lemma 3.1** (**Hessian of the NITP Objective**). *Let $s := u^\top v$ be the cosine alignment. The Hessian of $\mathcal{L}_{\mathrm{NITP}}$ with respect to $h$ is given by:*

$$H_{\mathrm{NITP}}(h) = \frac{1}{r^2} \left[ s(I - uu^\top) + (uA^\top + Au^\top) \right], \quad (5)$$

*where $A = v - su$ is the tangential difference vector. Near convergence, where the representation aligns with the target ($s \to 1$ and implies $A \to 0$), the Hessian simplifies to:*

$$H_{\mathrm{NITP}}(h) \approx \frac{1}{r^2} \underbrace{(I - uu^\top)}_{P_{\perp u}}. \quad (6)$$

*Here, $P_{\perp u}$ denotes the projection matrix onto the subspace orthogonal to $h$ (the tangent space of the hypersphere).*

*Proof.* See Appendix A.

**Remark: necessity of semantic targets.** The curvature guarantee relies on high alignment ($s \to 1$). This justifies the choice of *predictable* semantic targets derived from shallow layers over arbitrary vectors, which would yield weak alignment ($s \approx 0$). Empirically, we observe that the NITP loss, *i.e.* $1 - s$ consistently converges below $0.1$ (*e.g.*, $0.04$ for the 9B MoE), confirming that the optimization stably operates in the region where the Hessian is positive definite and geometric regularization is effective.

**Analysis of spectral properties.** Lemma 3.1 reveals two critical properties of NITP:

- **Radial null space:** $u^\top H_{\mathrm{NITP}} u = 0$. NITP imposes no curvature on the vector norm, allowing the NTP objective to freely adjust the scale ($r$)—for instance, to optimize softmax temperature—without interference.

- **Angular spectral lifting:** For any semantic deviation vector $w$ orthogonal to the current state ($w \perp u$), we have strictly positive curvature:

$$w^\top H_{\mathrm{NITP}} w \approx \frac{1}{r^2} \|w\|_2^2 > 0. \quad (7)$$

**Theorem 3.2** (**Null Space Mitigation in Semantic Subspace**). *Consider the total loss Hessian $H_{\mathrm{total}} = H_{\mathrm{NTP}} + \lambda H_{\mathrm{NITP}}$. Let $\mathcal{N}_{\mathrm{sem}}$ be the semantic null space of NTP, defined as the set of directions $w \perp u$ such that $w^\top H_{\mathrm{NTP}} w \approx 0$. With the addition of NITP, for any $w \in \mathcal{N}_{\mathrm{sem}}$, the curvature becomes strict:*

$$w^\top H_{\mathrm{total}} w \approx 0 + \frac{\lambda}{r^2} \|w\|_2^2 > 0. \quad (8)$$

See detailed analysis at Appendix A.

**Interpretation.** Theorem 3.2 demonstrates that NITP performs spectral lifting on the angular manifold. While standard NTP leaves vast degrees of freedom orthogonal to the target logit unpenalized, NITP effectively regularizes these under-constrained directions by introducing positive curvature. This mitigates the "flat valleys" in the optimization landscape responsible for semantic drift, forcing the model to maintain robust representations aligned with the semantic anchors derived from shallow layers.

## 4. Experiments

In this section, we conduct a comprehensive empirical evaluation to assess the effectiveness of Next Implicit Token Prediction (NITP) in terms of downstream performance. We train both dense and mixture-of-experts models from scratch, covering model scales from hundreds of millions to billions of parameters, and assess performance on a broad suite of knowledge and reasoning benchmarks. Additionally, we conduct ablation studies to examine the effect of key design choices in NITP and to verify its effectiveness.

### 4.1. Experiment Setups

**Training settings.** We evaluate NITP on both Mixture-of-Experts (MoE) (Shazeer et al., 2017) and dense language models to assess its generality across model architectures and scales. For MoE models, we adopt the DeepSeek-V2 (Liu et al., 2024a) architecture employing 144 experts with top-8 routing, where each expert is implemented as a SwiGLU-based (Shazeer, 2020) FFN. We scale this architecture up to a total parameter count of 9B, corresponding to

*Table 1.* **Main results.** Performance comparison between NTP and NITP. We categorize benchmarks into **Knowledge & Language** (5 tasks) and **Reasoning & Problem Solving** (8 tasks). Best results are **bolded**. 'A' denotes the number of activated parameters per token, and the number before 'A' represents the total parameters in MoE. **Abbreviations:** M-Pro denotes MMLU-Pro, LAMB. denotes LAMBADA, CSQA denotes CommonsenseQA, COPA denotes Balanced COPA.

| Model | Method | Knowledge & Language Understanding | | | | | Reasoning & Problem Solving | | | | | | | | Avg. |
|---|---|---|---|---|---|---|---|---|---|---|---|---|---|---|---|
| | | MMLU | M-Pro | C-Eval | Xiezhi | LAMB. | BBH | ARC-C | CSQA | COPA | C3 | AGIEval | GSM8k | LCBench | |
| 1.9bA0.3b | NTP | 31.05 | 7.14 | 32.31 | **32.13** | 50.36 | 17.70 | 24.74 | 25.38 | 61.20 | **32.21** | 24.57 | 6.06 | 1.56 | 26.65 |
| | **NITP** | **31.68** | **7.47** | **33.16** | 31.22 | **50.52** | **19.11** | **29.20** | **26.61** | **62.90** | 29.69 | **26.69** | **6.44** | **2.26** | **27.46** |
| 3bA0.5b | NTP | 34.60 | 11.00 | **33.53** | 36.53 | **55.45** | 21.92 | 32.65 | 34.15 | **65.60** | 39.06 | 26.67 | 12.81 | 3.82 | 31.37 |
| | **NITP** | **37.37** | **12.29** | 33.12 | **39.61** | **55.45** | **26.14** | **37.11** | **37.92** | **65.60** | **44.38** | **27.76** | **14.40** | **4.17** | **33.49** |
| 9bA1b | NTP | 43.71 | 15.29 | 38.97 | 45.59 | 62.32 | 28.07 | 51.20 | 45.70 | 68.10 | 56.65 | **30.91** | 30.09 | 6.95 | 40.27 |
| | **NITP** | **46.14** | **21.00** | **40.72** | **49.10** | **64.49** | **28.67** | **53.95** | **49.96** | **70.70** | **63.01** | 29.97 | **32.52** | **8.00** | **42.94** |

approximately 1B activated parameters per token. For dense models, we conduct experiments on model sizes ranging from 0.5B to 3B parameters. All models are trained from scratch with token budgets scaled according to their parameters, and within each comparison NTP and NITP share an identical token budget to ensure fairness. Specifically, the 9B MoE model is trained on a large-scale corpus of high-quality tokens comprising a diverse mix of English, Chinese, code, mathematics, and reasoning data. The context length is fixed at 8192 tokens unless stated otherwise. Comprehensive training details are provided in Appendix B.

**Evaluation.** We conduct comprehensive few-shot evaluations on a diverse suite of benchmarks to assess the effectiveness of NITP relative to the standard NTP objective. We categorize these benchmarks into two primary domains: **1) Knowledge & Language Understanding:** To evaluate broad world knowledge and multi-domain comprehension, we utilize English benchmarks MMLU (Hendrycks et al., 2020) and MMLU-Pro (Wang et al., 2024), alongside Chinese datasets C-Eval (Huang et al., 2023) and Xiezhi (Gu et al., 2024b). We additionally include LAMBADA (Paperno et al., 2016) to test long-range dependency modeling. **2) Reasoning & Problem Solving:** To examine logical deduction and complex problem-solving, we employ ARC-Challenge (Clark et al., 2018) for scientific reasoning, together with CommonsenseQA (Talmor et al., 2019) and Balanced COPA (Kavumba et al., 2019) for commonsense reasoning. We also include BBH (Suzgun et al., 2023) and C3 (Sun et al., 2020a) to evaluate challenging multi-step inference and reading comprehension. Furthermore, we assess rigorous application skills using AGIEval (Zhong et al., 2024) on standardized exams, GSM8k (Cobbe et al., 2021) on mathematics, and OpenCompass LCBench (Contributors, 2023) (a collection of programming problems from Leetcode weekly competitions) on code generation.

### 4.2. Main Results

**Evaluation on MoE models.** We mainly evaluate the performance of the proposed NITP against the NTP baseline

across three Mixture-of-Experts (MoE) scales: 1.9B (0.3B activated), 3B (0.5B activated), and 9B (1B activated). The models were pre-trained from scratch with increasing token budgets proportional to their scale. The quantitative results are reported in Table 1. The results demonstrate that NITP consistently outperforms standard NTP across all model sizes, underscoring the method's scalability. Specifically, NITP yields average score improvements of 0.8, 2.1, and 2.7 points for the 1.9B, 3B, and 9B models, respectively. For instance, on the 9B model, NITP achieves a 5.71% absolute improvement on MMLU-Pro (rising from 15.29% to 21.00%), a benchmark specifically designed to test robust reasoning beyond simple knowledge retrieval. Similarly, we observe remarkable boosts in reading comprehension and commonsense reasoning, with C3 scores increasing by 6.36% (56.65% to 63.01%) and CommonsenseQA by 4.26% (45.70% to 49.96%). Also, on symbolic and math tasks like ARC-C and GSM8k, NITP maintains a consistent lead.

**Evaluation on dense models.** To assess the generality of NITP beyond MoE architectures, we evaluate dense models ranging from 0.5B to 3B parameters on seven representative benchmarks, including MMLU, C-Eval, BBH, ARC-C, C3, AGIEval, and LCBench. The quantitative results are reported in Table 2. The results demonstrate that NITP is architecture-agnostic, consistently outperforming the NTP baseline across all model scales. On the 0.5B model, NITP induces notable improvements; specifically, C3 scores surge by 3.89 points, and the average score increases from 24.42% to 25.44%. For the 2B model, NITP improves the average score by 1.79 points (31.91% to 33.70%), with notable gains on C3 (+4.16) and MMLU (+2.18). Similarly, on the 3B model, NITP achieves an average improvement of 1.35 points, accompanied by substantial gains on AGIEval and C3. Overall, these results confirm that NITP consistently enhances representation quality across diverse tasks, independent of model architecture such as MoE or dense.

Collectively, these empirical results validate the core design philosophy of NITP. By augmenting discrete prediction with semantic-rich continuous supervision, NITP effectively

*Table 2.* **Results for dense models.** Performance comparison between NTP and NITP on seven representative benchmarks. Best results are **bolded**. **Avg.** denotes the arithmetic mean across all benchmarks.

| Model | Method | MMLU | C-Eval | BBH | ARC-C | C3 | AGIEval | LCBench | **Avg.** |
|-------|--------|------|--------|-----|-------|-----|---------|---------|----------|
| 0.5B | NTP | 30.59 | **33.72** | 18.96 | 31.95 | 28.21 | **27.01** | 0.52 | 24.42 |
| | NITP | **31.01** | 32.78 | **19.92** | **34.71** | **32.10** | 26.15 | **1.39** | **25.44** |
| 2B | NTP | 37.96 | 35.67 | 26.59 | 39.52 | 49.26 | 30.51 | **3.83** | 31.91 |
| | **NITP** | **40.14** | **37.81** | **27.33** | **41.92** | **53.42** | **31.66** | 3.65 | **33.70** |
| 3B | NTP | 43.54 | 39.17 | **31.25** | 51.20 | 59.01 | 31.77 | 4.35 | 37.18 |
| | **NITP** | **44.95** | **40.14** | 29.40 | **51.55** | **63.67** | **35.11** | **4.87** | **38.53** |

constrains the latent geometry, preventing hidden states from drifting into degenerate configurations. This translates into consistent performance improvements across diverse architectures (MoE and Dense). Crucially, the substantial gains observed across the evaluated MoE and dense scales further support the scalability of NITP. Beyond downstream benchmark accuracy, we directly evaluate the utility of the learned hidden states as frozen sentence representations in Appendix D, where NITP improves 23 out of 25 MTEB tasks while preserving validation cross-entropy loss.

**Training efficiency.** We analyze the computational efficiency of NITP by comparing its complexity with the standard NTP baseline. For a MoE model, the training cost is dominated by $L$ layers of backbone blocks and the unembedding projection, scaling as $O(L(d^2 + kdd_e) + Vd)$, where $k$ is the number of activated experts, $d_e$ is the expert dimension, and $V$ is the vocabulary size. In contrast, the NITP objective introduces a single projection head and a cosine loss, with a total complexity of $O(d^2)$. Crucially, since the implicit targets are derived from existing intermediate activations and decoupled via a stop-gradient operator, NITP incurs no additional backbone forward passes or backward propagation through the early layers. Given that $L \gg 1$, the relative complexity ensures that the extra overhead is inherently marginal. Numerical instantiation on our 9B MoE model ($d = 1280, L = 24$) confirms this advantage. **The additional training FLOPs from both forward and backward passes account for approximately 2% of the total computation.** This negligible cost ensures that NITP serves as a highly scalable supervision objective without sacrificing training throughput. See details in Appendix F.

### 4.3. Ablation Study

We conduct ablation studies to understand how each component contributes to its effectiveness. In particular, we focus on the following aspects: 1) how to construct the implicit prediction target, including the choice of target layer and temporal shift. 2) how to design a stable and effective auxiliary objective, including the loss function and its role beyond generic regularization. Unless otherwise specified,

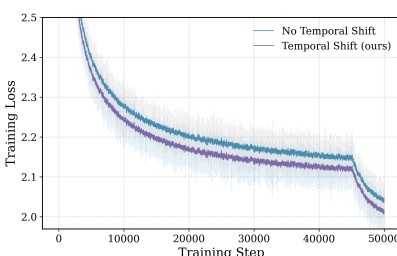

*Figure 3.* Loss comparison between whether temporal shift or not.

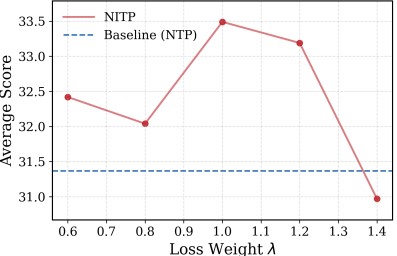

*Figure 4.* Average performance under different NITP loss.

all ablation experiments are conducted on a 3B-parameter MoE model in Table 4.

**Importance of temporal shift.** To distinguish NITP from static layer-wise alignment, we ablate the temporal structure of the implicit prediction objective. Specifically, we compare predicting the semantic representation of the next token, *i.e.* the next implicit token, with aligning semantic representations at the current time step. As shown in Table 3 and Figure 3, predicting the next implicit token significantly outperforms same-position alignment, while the latter leads to terrible loss and performance degradation. Notably, same-position alignment can achieve a very low alignment loss (*e.g.*, 0.03), yet results in substantially worse downstream performance. This indicates that simply regularizing representations or minimizing alignment loss is insufficient. Without the autoregressive temporal shift, **the objective fails to provide a meaningful predictive signal of the next token and can collapse into trivial solutions.** These results highlight that the effectiveness of NITP crucially depends on predicting the future semantic representation, rather than

*Table 3.* **Ablation studies.** We evaluate the impact of key design choices in NITP using the 3B MoE model, including the target layer selection, temporal shift strategy, loss function, and regularization. The default NITP configuration (shallow $L_4$, next-token target, cosine loss) achieves the best performance. $L_4$ denotes selecting the 4th layer out of 17 layers as the target layer.

| Ablation Factor | Setting | MMLU | MMLU-Pro | CSQA | BBH | LCBench | Avg. |
|---|---|---|---|---|---|---|---|
| Baseline: NTP | | 34.60 | 11.00 | 34.15 | 21.92 | 3.82 | 21.10 |
| Target Layer | Deep ($L_{14}$) | 35.79 | 10.43 | **38.90** | 23.25 | 2.43 | 22.16 |
| | Middle ($L_8$) | 35.33 | 11.57 | 34.72 | 22.07 | 2.43 | 21.22 |
| | **Shallow** ($L_4$) | **37.37** | **12.29** | 37.92 | **26.14** | **4.17** | **23.58** |
| Temporal Shift | Current-step ($t \to t$) | 33.09 | 8.14 | 29.15 | 20.96 | 2.43 | 18.75 |
| | **Next-token** ($t \to t + 1$) | **37.37** | **12.29** | **37.92** | **26.14** | **4.17** | **23.58** |
| Loss Function | MSE Loss | 32.77 | 10.29 | 30.38 | 21.55 | 1.91 | 19.38 |
| | Smooth L1 | **39.72** | 11.43 | 37.51 | 24.74 | 2.78 | 23.24 |
| | Distillation (KL) | 34.67 | 11.14 | **38.08** | **26.88** | 3.65 | 22.88 |
| | **Cosine Similarity** | 37.37 | **12.29** | 37.92 | 26.14 | **4.17** | **23.58** |
| Regularization | Generic Cosine Reg. | 34.45 | 10.14 | 33.25 | 22.29 | 3.82 | 20.79 |
| | **NITP** | **37.37** | **12.29** | **37.92** | **26.14** | **4.17** | **23.58** |

arbitrarily aligning hidden states across layers.

**Loss function.** We compare the Cosine Similarity against regression-based losses (MSE, Smooth-$\ell_1$) and KL Divergence in Table 3. The results highlight significant stability differences. MSE leads to catastrophic optimization divergence that loss is observed to diverge temporarily during training, as its quadratic penalty amplifies the inherent scale mismatch between layers, triggering destructive gradient spikes. Though Smooth-$\ell_1$ maintains stability via implicit gradient clipping, it still remains unknown for larger-scale training due to its hard alignment form. KL Divergence underperforms as it treats continuous feature vectors as distributions, causing geometric distortion. Ultimately, Cosine Similarity proves to be the superior choice. Empirically, it is the most stable objective without any divergence issues and consistently delivers superior performance across tasks.

**Target layer selection.** A key design choice in NITP is the use of shallow-layer representations as prediction targets (implicit tokens) under the temporal-shifted prediction objective. To validate this choice, we vary the source layer of the implicit tokens across shallow, middle, and deep layers of the model. The results in Table 3 show that using shallow-layer targets consistently outperforms the alternatives. We attribute this behavior to the fact that shallow layers preserve richer lexical and local semantic structure, whereas deeper layers are more specialized for token discrimination and exhibit worse semantics (Skean et al., 2025). Among shallow layers, we further find that selecting an early layer around 20% of the total depth yields the best performance (see Tables 3 and 6), which is consistent with prior findings that semantic richness peaks in specific early layers (Skean et al., 2025; Liu et al., 2024c) and aligns with our analysis in Section 3.3.

**Is NITP merely a hidden state regularization?** To examine whether NITP reduces to generic hidden-state regular-

ization, we compare it with a cosine-based baseline inspired by (Gao et al., 2019). Specifically, this baseline applies a cosine similarity penalty between hidden states of different tokens within sequences, encouraging them to be less aligned. While this regularization constrains the geometry of hidden representations, it does not provide a prediction target or token-level supervision. In contrast, NITP introduces a prediction-aligned auxiliary objective by training the final hidden state to predict the next implicit token, which encodes context-dependent information about the upcoming token. As shown in Table 3, we find that the cosine regularization baseline yields no improvements, whereas NITP consistently improves performance, indicating that the gains stem from prediction-aligned semantic supervision rather than merely representation regularization.

**NITP loss weight.** We study the effect of the NITP loss weight by sweeping $\lambda$ in Equation (4). As shown in Figure 4, performance consistently peaks around $\lambda = 1.0$, which emerges as the most reliable choice across settings. While the exact optimum may vary slightly across model configurations, strong performance is consistently observed for values around 1.0, indicating low sensitivity within this range. The specific loss weights used for different models are summarized in Table 4.

## 5. Conclusion

We propose Next Implicit Token Prediction (NITP), a novel pre-training objective that augments standard next-token prediction with continuous supervision in the representation space. Motivated by the observation that NTP leaves hidden representations under-constrained and prone to geometric degeneration, NITP requires the model to predict the implicit semantic representation of the next token using shallow-layer states as self-supervised targets, which we term as next implicit token. This objective explicitly con-

strains the free degrees of freedom ignored by token-level supervision and prevents representation degeneration. We provide a theoretical analysis showing that NITP mitigates the optimization null space, thereby regularizing representation geometry. Experiments across both dense and MoE models demonstrate that NITP consistently improves downstream performance with minimal computational overhead.

**Limitations.** NITP introduces additional hyperparameters, including the target layer and NITP loss weight. While our experiments indicate that these choices are highly stable across models, further validation on more models is required to fully confirm this robustness.

## Acknowledgements

This work was in part supported by Scientific Research Innovation Capability Support Project for Young Faculty (U40) of the Ministry of Education of China (SRICSPYF-ZY2025019) and Xiaohongshu Inc.

## Impact Statement

This work introduces a representation-level auxiliary objective for language model pre-training, with the goal of improving optimization stability and the geometric structure of hidden representations. The proposed method modifies only the training objective and does not involve new data sources, model architectures, or deployment settings. As a result, the societal and ethical impacts of this work are expected to be aligned with those of existing large language models trained using standard next-token prediction. Although improved representation quality and training efficiency may lead to better downstream performance, this does not fundamentally change the usage scenarios or risk profile of current language modeling systems.

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

## A. Detailed Proofs for NITP Geometry

In this section, we provide the detailed derivation of the gradient and Hessian matrix for the Next Implicit Token Prediction (NITP) objective. We further analyze the spectral properties of the Hessian to validate the geometric regularization claims made in Section 3.5, providing the formal proof for Lemma 3.1 and Theorem 3.2.

### A.1. Notation and Preliminaries

Consistent with the setup in Section 3.5, let $h \in \mathbb{R}^d$ denote the representation used for the implicit prediction loss. We treat $h$ as the projected state to isolate the geometric incentives of the objective function. Let $z \in \mathbb{R}^d$ denote the fixed implicit target derived from shallow layers. We define the following variables:

$$r := \|h\|_2, \qquad\qquad u := \frac{h}{r},$$

$$v := \frac{z}{\|z\|_2}, \qquad\qquad s := u^\top v = \cos(h, z).$$

We assume $r > 0$. The tangential difference vector is defined as $A := v - su$. Note that $A$ is orthogonal to $u$:

$$u^\top A = u^\top (v - su) = u^\top v - s(u^\top u) = s - s = 0. \tag{9}$$

The NITP objective function is:

$$\mathcal{L}(h) = 1 - \cos(h, z) = 1 - \frac{h^\top z}{\|h\|_2 \|z\|_2} = 1 - u^\top v. \tag{10}$$

### A.2. Gradient Derivation

First, recall the Jacobian of the normalization map $u(h) = h/r$:

$$J_u(h) = \frac{\partial u}{\partial h} = \frac{1}{r}(I - uu^\top). \tag{11}$$

Let $f(h) = u^\top v$ be the cosine similarity. By the chain rule:

$$\begin{aligned}
\nabla_h f(h) &= J_u(h)^\top v \\
&= \frac{1}{r}(I - uu^\top)v \\
&= \frac{1}{r}(v - u(u^\top v)) \\
&= \frac{1}{r}(v - su) = \frac{A}{r}.
\end{aligned} \tag{12}$$

Thus, the gradient of the loss $\mathcal{L}(h) = 1 - f(h)$ is:

$$\nabla_h \mathcal{L}(h) = -\frac{A}{r}. \tag{13}$$

### A.3. Hessian Derivation (Proof of Lemma 3.1)

The Hessian is the Jacobian of the gradient vector. We differentiate $\nabla_h \mathcal{L}(h) = -r^{-1}A$ with respect to $h$:

$$H_{\text{NITP}}(h) = \nabla_h \left( -\frac{A}{r} \right) = -\left( \frac{1}{r}\nabla_h A + A(\nabla_h r^{-1})^\top \right). \tag{14}$$

We compute the two terms separately.

**Term 1: Derivative of $r^{-1}$.** Using $\nabla_h r = u$:

$$\nabla_h(r^{-1}) = -\frac{1}{r^2}\nabla_h r = -\frac{u}{r^2}. \tag{15}$$

Thus, the second part of the product rule is:

$$A(\nabla_h r^{-1})^\top = A\left(-\frac{u^\top}{r^2}\right) = -\frac{1}{r^2}Au^\top. \tag{16}$$

**Term 2: Derivative of $A$.** Recall $A = v - su$. Since $v$ is constant:

$$\nabla_h A = -\nabla_h(su) = -\left(u(\nabla_h s)^\top + s\nabla_h u\right). \tag{17}$$

Substituting $\nabla_h s = \nabla_h f = A/r$ and $\nabla_h u = \frac{1}{r}(I - uu^\top)$:

$$\nabla_h A = -\left(u\left(\frac{A}{r}\right)^\top + s\frac{1}{r}(I - uu^\top)\right)$$

$$= -\frac{1}{r}\left(uA^\top + s(I - uu^\top)\right). \tag{18}$$

**Combining the Terms.** Substituting back into the expression for $H_{\text{NITP}}$:

$$H_{\text{NITP}}(h) = -\left[\frac{1}{r}\left(-\frac{1}{r}(uA^\top + s(I - uu^\top))\right) + \left(-\frac{1}{r^2}Au^\top\right)\right]$$

$$= \frac{1}{r^2}(uA^\top + s(I - uu^\top)) + \frac{1}{r^2}Au^\top$$

$$= \frac{1}{r^2}\left[s(I - uu^\top) + (uA^\top + Au^\top)\right]. \tag{19}$$

This provides the exact Hessian form.

**Approximation near convergence.** As the training progresses and the prediction aligns with the target, we have $s \to 1$. Consequently, the difference vector $A = v - su$ approaches zero ($\|A\| \to 0$). Under this condition, the term $(uA^\top + Au^\top)$ vanishes. The Hessian simplifies to:

$$H_{\text{NITP}}(h) \approx \frac{1}{r^2}(I - uu^\top) = \frac{1}{r^2}P_{\perp u}. \tag{20}$$

This confirms the simplified form presented in Lemma 3.1. □

### A.4. Analysis of Spectral Properties

We now verify the properties stated in the main text regarding the null space and spectral lifting.

**Property 1: Radial Null Space.** We examine the curvature along the radial direction $u$. We compute the quadratic form $u^\top H_{\text{NITP}}u$:

$$u^\top H_{\text{NITP}}u = \frac{1}{r^2}\left[s\underbrace{u^\top(I - uu^\top)u}_{0} + \underbrace{u^\top(uA^\top)u}_{u^\top u(A^\top u)} + \underbrace{u^\top(Au^\top)u}_{u^\top A(u^\top u)}\right]. \tag{21}$$

Since $I - uu^\top$ projects onto the orthogonal complement of $u$, the first term is 0. For the remaining terms, recall that $A \perp u$, so $A^\top u = 0$ and $u^\top A = 0$.

$$u^\top H_{\text{NITP}}u = \frac{1}{r^2}[0 + 1 \cdot 0 + 0 \cdot 1] = 0. \tag{22}$$

This confirms that the loss function imposes zero curvature (is locally linear or flat) with respect to changes in the vector norm.

**Property 2: Angular Spectral Lifting.** Consider a perturbation vector $w \in \mathbb{R}^d$ such that $w \perp u$ (representing a purely semantic change). We compute $w^\top H_{\text{NITP}}w$:

$$w^\top H_{\text{NITP}}w = \frac{1}{r^2}\left[sw^\top(I - uu^\top)w + w^\top(uA^\top + Au^\top)w\right]. \tag{23}$$

Since $w \perp u$: 1. $w^\top (I - uu^\top)w = w^\top w - (w^\top u)^2 = \|w\|_2^2$. 2. $w^\top u A^\top w = (w^\top u)(A^\top w) = 0$. 3. $w^\top A u^\top w = (w^\top A)(u^\top w) = 0$.

Thus, the curvature in the tangential direction is:

$$w^\top H_{\text{NITP}} w = \frac{s}{r^2} \|w\|_2^2. \tag{24}$$

Assuming reasonable alignment during training ($s = \cos(h, z) > 0$), the Hessian is strictly positive definite on the subspace orthogonal to $h$.

### A.5. Proof of Theorem 3.2 (Null Space Mitigation)

We consider the total loss Hessian $H_{\text{total}} = H_{\text{NTP}} + \lambda H_{\text{NITP}}$.

**Existence of null space.** As analyzed in Section 3.2, although the NTP objective theoretically involves all vocabulary items via the Softmax normalization, the gradient signal is empirically dominated by the target token and a sparse set of high-probability contenders. Consequently, the curvature of $H_{\text{NTP}}$ is concentrated within this low-dimensional subspace, leaving the vast orthogonal complement effectively unconstrained (rank-deficient).

Based on this, let $\mathcal{N}_{\text{sem}}$ be the *semantic null space* of the standard NTP objective, defined as the set of tangential directions $w \perp u$ that lie in this unconstrained subspace ($w^\top H_{\text{NTP}} w \approx 0$).

For any such degenerate direction $w \in \mathcal{N}_{\text{sem}}$, we compute the quadratic form of the total Hessian:

$$\begin{aligned} w^\top H_{\text{total}} w &= w^\top (H_{\text{NTP}} + \lambda H_{\text{NITP}})w \\ &= \underbrace{w^\top H_{\text{NTP}} w}_{\approx 0} + \lambda w^\top H_{\text{NITP}} w. \end{aligned} \tag{25}$$

Substituting the result from Property 2 derived above ($w^\top H_{\text{NITP}} w = \frac{s}{r^2} \|w\|_2^2$):

$$w^\top H_{\text{total}} w \approx 0 + \lambda \left( \frac{s}{r^2} \|w\|_2^2 \right). \tag{26}$$

Under the empirical observation that NITP converges to high alignment ($s \to 1$), we have $s > 0$. Given $\lambda > 0$, it follows that:

$$w^\top H_{\text{total}} w > 0 \quad \text{for all } w \in \mathcal{N}_{\text{sem}}, w \neq 0. \tag{27}$$

This proves that the addition of NITP mitigates the semantic null space by ensuring strictly positive curvature in these previously unconstrained directions. $\square$

## B. Experiment Details

**Hyperparameter settings.** Table 4 summarizes the hyperparameter configurations used for pretraining both MoE and Dense models. Across all model scales, we adopt a largely consistent training recipe to ensure fair comparisons between standard NTP and NITP. All models are optimized using AdamW with momentum coefficients $(\beta_1, \beta_2) = (0.9, 0.95)$, weight decay set to 0.1, and global gradient clipping at 1.0. We employ a warmup–stable–decay (WSD) learning rate schedule (Wen et al., 2024) with 2,000 warmup steps and a fixed decay ratio of 0.2. The learning rate and global batch size are scaled according to model size, while the context length is fixed to 8192 tokens for all experiments.

**Model architectures.** For MoE models, the backbone consists of a shallow dense stage followed by multiple MoE layers, with the number of MoE layers increasing with model capacity. Each MoE layer contains 144 routed experts and one shared expert, using top-8 routing throughout. To control the overall parameter budget, the expert feed-forward hidden size is adjusted across scales, while the dense FFN size is kept fixed. All MoE models use multi-head attention with 8 query heads and 4 key–value heads, with head dimensions scaled proportionally to the hidden size.

Dense models follow a standard Transformer architecture with 24–28 layers depending on model size. They use a larger number of attention heads and higher-dimensional FFNs to match the overall parameter scale of the MoE counterparts. For both MoE and Dense models, the NITP target layer is chosen from intermediate depths, and the NITP loss weight is set close to 1.0, with minor adjustments for larger models. All models are pretrained on large-scale tokenized corpora using the

*Table 4.* Hyper-parameters for MoE and Dense model pretraining.

| Parameters | | MoE Models | | | Dense Models | | |
|---|---|---|---|---|---|---|---|
| | | **1.9bA0.3b** | **3bA0.5b** | **9bA1b** | **0.5B** | **2B** | **3B** |
| Training | lr-schedule | WSD | WSD | WSD | WSD | WSD | WSD |
| | lr | 6.0e-4 | 8.7e-4 | 4.2e-4 | 4.2e-4 | 4.2e-4 | 4.2e-4 |
| | warmup steps | 2,000 | 2,000 | 2,000 | 2,000 | 2,000 | 2,000 |
| | decay-ratio | 0.2 | 0.2 | 0.2 | 0.2 | 0.2 | 0.2 |
| | AdamW-$\beta$ | (0.9, 0.95) | (0.9, 0.95) | (0.9, 0.95) | (0.9, 0.95) | (0.9, 0.95) | (0.9, 0.95) |
| | weight-decay | 0.1 | 0.1 | 0.1 | 0.1 | 0.1 | 0.1 |
| | grad_clip | 1.0 | 1.0 | 1.0 | 1.0 | 1.0 | 1.0 |
| | global batch size | 256 | 512 | 1024 | 512 | 1024 | 1024 |
| Model | hidden dim. | 512 | 768 | 1280 | 896 | 1792 | 2560 |
| | #dense layers | 1 | 1 | 1 | 24 | 28 | 28 |
| | #moe layer | 15 | 16 | 23 | - | - | - |
| | head dim. | 64 | 96 | 160 | 64 | 128 | 128 |
| | #q heads | 8 | 8 | 8 | 14 | 14 | 20 |
| | #kv heads | 4 | 4 | 4 | 2 | 2 | 4 |
| | context-length | 8192 | 8192 | 8192 | 8192 | 8192 | 8192 |
| | #routed experts | 144 | 144 | 144 | - | - | - |
| | #shared experts | 1 | 1 | 1 | - | - | - |
| | topK route | 8 | 8 | 8 | - | - | - |
| | dense FFN size | 10944 | 10944 | 10944 | 4864 | 10752 | 10240 |
| | MoE FFN size | 496 | 544 | 640 | - | - | - |
| | shared exp. size | 496 | 544 | 640 | - | - | - |
| | NITP target layer | 3/16 | 4/17 | 5/24 | 5/24 | 6/28 | 6/28 |
| | NITP loss weight | 1.0 | 1.0 | 0.8 | 1.0 | 1.0 | 0.8 |
| Data | tokenizer | qwen2 | qwen2 | qwen2 | qwen2 | qwen2 | qwen2 |

Qwen2 tokenizer. The total number of training tokens scales with model size and architecture, with each NTP and NITP pair trained under an identical token budget to ensure fair comparison.

For the average downstream performance shown in the bottom row of Figure 1, we average over four representative benchmarks: MMLU, C3, C-Eval, and ARC-Challenge.

## C. Analysis of Representation Geometry Dynamics

We provide a detailed analysis of the geometric phenomena observed in Figure 1, focusing on the joint behavior of *effective rank* and *average cosine similarity* during training.

**Effective rank dynamics.** We periodically measure the effective rank of the last-layer hidden states during training by collecting all valid token representations within a mini-batch, applying mean-centering, and computing the entropy-based effective rank from the eigenvalue spectrum of the empirical covariance matrix. Measurements are reported every fixed number of training steps to track long-term geometric trends.

As shown in Figure 1(a), where the background indicates the variance of effective rank, standard NTP exhibits a rapid and monotonic collapse of effective rank, converging to a low-variance regime early in training. This behavior indicates that the learned hidden states concentrate on a small, fixed subspace, regardless of contextual variation. Such convergence reflects a degenerate equilibrium: once the model discovers a narrow anisotropic cone that suffices for next-token prediction, no gradient signal encourages re-expansion along semantically meaningful but task-irrelevant directions.

In contrast, NITP maintains a substantially higher effective rank with noticeably larger variance over training. Importantly, this variance should not be interpreted as optimization instability. Rather, it is a direct consequence of mitigating the *semantic null space* inherent to NTP. By explicitly supervising the prediction of the next token in latent space, NITP introduces positive curvature along directions that are otherwise unconstrained under likelihood-based training. As a result, the model

dynamically activates different subsets of the representation space depending on contextual demands, leading to controlled fluctuations in effective rank.

**Cosine similarity and anisotropy.** To quantify global anisotropy, we report the average cosine similarity between randomly sampled pairs of token representations from the last layer. At each reporting step, a fixed number of token pairs is sampled across sentences, and cosine similarity is computed after $\ell_2$ normalization.

As shown in Figure 1(b), under NTP the average cosine similarity increases steadily throughout training, signaling growing anisotropy and alignment toward a common dominant direction. This behavior is consistent with prior observations of representation degeneration, where embeddings become increasingly indistinguishable despite maintaining predictive accuracy. NITP alleviates this effect. Although cosine similarity still increases as training progresses, the overall magnitude is consistently lower, indicating that representations preserve greater angular diversity. This reduced anisotropy aligns with the higher effective rank observed under NITP, jointly suggesting that the representation space remains more expressive and better conditioned.

**Implications.** Taken together, these results clarify that NITP does not merely slow down representation collapse, but fundamentally alters the geometric equilibrium of training. While NTP permits hidden states to sacrifice expressiveness as long as token prediction remains correct, NITP enforces semantic consistency in latent space, preventing representations from collapsing into a static anisotropic cone. This geometric regularization provides a mechanistic explanation for the improved downstream generalization observed with NITP.

## D. Hidden-State Representation Quality

NITP is designed to improve the representations learned during pre-training, rather than merely to change final benchmark scores through the output head. While Appendix C characterizes this effect through representation geometry, we further examine whether the learned hidden states encode richer semantic information than the baseline.

**Hidden-state quality on MTEB.** We evaluate frozen sentence representations on MTEB v2.12.0 (Muennighoff et al., 2023). Following standard practice, for each example, we extract the last-layer hidden states from the 3B MoE models, mean-pool over valid tokens, and apply $\ell_2$ normalization. No fine-tuning, contrastive training, or task-specific adaptation is applied, so the evaluation serves as a direct probe of the representation quality induced by the pre-training objective. We evaluate 25 English MTEB tasks and exclude seven large-scale clustering and reranking tasks due to computational cost. According to the task types used in our evaluation pipeline, the remaining tasks are grouped into Classification (10 tasks), STS/Similarity (9 tasks), and Retrieval/Duplicate-detection (6 tasks).

The evaluated tasks are: *Classification*: AmazonCounterfactualClassification, AmazonReviewsClassification, Banking77Classification, EmotionClassification, MTOPDomainClassification, MTOPIntentClassification, MassiveIntentClassification, MassiveScenarioClassification, ToxicConversationsClassification, and TweetSentimentExtractionClassification; *STS/Similarity*: BIOSSES, SICK-R, STS12, STS13, STS14, STS15, STS16, STS17, and STSBenchmark; and *Retrieval/Duplicate-detection*: AskUbuntuDupQuestions, SciDocsRR, SprintDuplicateQuestions, StackOverflowDupQuestions, TwitterSemEval2015, and TwitterURLCorpus.

*Table 5.* **MTEB evaluation of hidden-state quality.** Scores are reported as percentages and macro-averaged over task-level main scores within each group. NITP improves representation utility on 23 out of 25 tasks using frozen last hidden states.

| Task Group | NTP | NITP | $\Delta$ |
|---|---|---|---|
| Classification (10) | 40.09 | 42.29 | +2.20 |
| STS / Similarity (9) | 35.66 | 38.58 | +2.93 |
| Retrieval / Dup. (6) | 43.19 | 44.83 | +1.64 |
| **Overall (25 tasks)** | **39.24** | **41.56** | **+2.33** |

As shown in Table 5, NITP improves the overall MTEB score from 39.24 to 41.56. The gains are consistent across task families, with improvements of +2.20 points on classification, +2.93 points on semantic textual similarity, and +1.64 points on retrieval/duplicate-detection tasks. At the individual task level, NITP improves 23 out of 25 tasks; the only regressions are small drops on AskUbuntuDupQuestions ($-0.15$ points) and ToxicConversationsClassification ($-0.88$ points). These

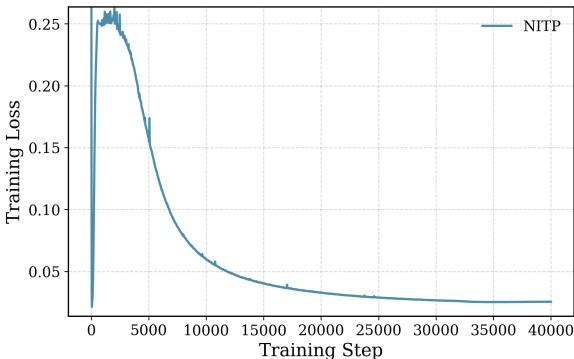

*Figure 5.* **Training dynamics of the NITP loss.** Evolution of $\mathcal{L}_{\text{NITP}}$ during pre-training, exhibiting a characteristic three-phase behavior: an initial collapse induced by random initialization, a transient hump caused by the emergence of structured shallow-layer targets, and a long-term stable convergence.

results indicate that NITP improves the transferability of the hidden states themselves, providing direct evidence for its role as a representation-level pre-training signal.

**Language modeling preservation.**  Finally, we verify that the stronger representations do not come from sacrificing the original language modeling objective. On the Pile (Gao et al., 2020) validation set, NITP achieves nearly identical validation cross-entropy loss to NTP across model configurations: 3B MoE obtains 2.006 vs. 2.006 (PPL 7.43 vs. 7.43), 9B MoE obtains 1.841 vs. 1.840 (PPL 6.30 vs. 6.30), and 3B Dense obtains 1.884 vs. 1.882 (PPL 6.58 vs. 6.57), where each pair reports NTP vs. NITP. This suggests that token-level cross-entropy alone does not fully characterize the quality of the learned hidden states: models with matched next-token likelihood can still differ substantially in representation geometry and transferability, consistent with our core claim. NITP therefore preserves validation loss under the original next-token prediction objective while learning hidden-state representations with better semantic transferability.

# E. Loss Curve of NITP

To further understand the optimization dynamics of Next Implicit Token Prediction, we monitor the trajectory of $\mathcal{L}_{\text{NITP}}$ throughout pre-training (Figure 5). The loss curve exhibits a characteristic three-phase evolution:

- **Phase I: random-initialization collapse.** In the very early stage of training, the NITP loss drops sharply from its initial value within the first few dozen steps. This behavior is primarily induced by random initialization: both the predictive state $h_t$ and the implicit target $z_{t+1}$ are nearly isotropic and weakly differentiated across tokens, causing their cosine similarity to rapidly concentrate around a common value. As a result, the NITP objective collapses quickly before any meaningful representation learning takes place.

- **Phase II: emergence of structured targets.** Following the initial drop, the loss exhibits a transient increase, forming a local peak (around 40–2k steps, *i.e.* warm-up phase). This behavior is consistent with the bottom-up convergence pattern observed in language models (Skean et al., 2025). As shallow layers converge faster, they begin to extract more structured and contextualized representations, transforming the implicit target $z_{t+1}$ from a near-constant signal into a more informative prediction target. This transition temporarily increases the difficulty of the NITP objective for the deeper predictive state $h_t$.

- **Phase III: steady convergence.** After the peak, the loss enters a prolonged and stable decline, indicating that deeper layers progressively adapt to the representational manifold defined by shallow layers. The final convergence to a low loss value (*e.g.*, below 0.05 for our 9B model) is consistent with NITP providing a stable and learnable auxiliary supervision signal throughout pre-training.

Overall, the observed hump in the loss curve suggests that the NITP objective functions not merely as a static regularizer, but as an adaptive predictive task whose difficulty evolves alongside the model's internal representations.

# F. Training FLOPs Analysis

This section provides a detailed analysis of the training FLOPs introduced by NITP, in comparison with the baseline Next-Token Prediction (NTP) objective. Following common practice (Kaplan et al., 2020; Hoffmann et al., 2022), we approximate total training FLOPs as approximately $6\times$ the number of parameters involved in forward computation, accounting for forward, backward, and parameter gradient updates.

## F.1. Baseline NTP FLOPs

We consider a transformer-based MoE model with model hidden size $d$ and vocabulary size $V$. Each layer consists of a self-attention module and a MoE block. Additionally, the unembedding layer projects the final hidden state to the vocabulary space. For each token:

- **Self-attention (GQA) (Ainslie et al., 2023):** Adopting Grouped Query Attention (GQA) with 8 query heads and 4 KV heads, the parameter count is reduced compared to standard Multi-Head Attention. The projections $(W_Q, W_K, W_V, W_O)$ contribute $\approx 3d^2$ parameters (where $W_K$ and $W_V$ are halved), resulting in $\approx 18d^2$ training FLOPs.

- **MoE MLP (SwiGLU (Shazeer, 2020)):** Each expert uses three projections $(d \times d_e)$. For $k$ activated experts, the training FLOPs are $\approx 6 \times (3kdd_e) = 18kdd_e$.

- **Unembedding Layer:** The final projection to the large vocabulary space $(d \times V)$ contributes significant computation. The training FLOPs are $\approx 6Vd$. Note that the input embedding layer is excluded as it involves only lookup operations (0 FLOPs).

The total baseline training FLOPs per token (summing over $L$ layers and the output head) are:

$$\text{FLOPs}_{\text{NTP}} \approx L \times (18d^2 + 18kdd_e) + 6Vd. \tag{28}$$

## F.2. Additional FLOPs Introduced by NITP

NITP introduces a projection head and a cosine loss. Crucially, extracting the target $z_{t+1}$ incurs no cost due to the stop-gradient.

**Projection head.** We employ a SwiGLU projection head with an intermediate dimension of $4d$. It consists of three linear transformations ($d \to 4d$, $d \to 4d$, and $4d \to d$), resulting in $12d^2$ parameters. The training FLOPs are:

$$\text{FLOPs}_{\text{proj}} \approx 6 \times 12d^2 = 72d^2. \tag{29}$$

**Cosine similarity loss.** The computation involves dot products and norms ($\approx 6d$ operations). Including the backward pass:

$$\text{FLOPs}_{\text{cos}} \approx 3 \times 6d = 18d. \tag{30}$$

## F.3. Numerical Instantiation: 9B MoE Model

Using the 9B MoE model parameters specified in Table 4 ($d = 1280$, $d_e = 640$, $k = 9$, $L = 24$, $V = 152064$), we estimate the FLOPs as follows:

- **NITP Overhead (Total):** $72d^2 + 18d \approx 72 \times 1280^2 \approx 1.18 \times 10^8$ FLOPs.

- **Baseline (Backbone Layers):** $24 \times (18 \times 1280^2 + 18 \times 9 \times 1280 \times 640) \approx 24 \times 1.62 \times 10^8 \approx 3.89 \times 10^9$ FLOPs.

- **Baseline (Unembedding):** $6 \times 152064 \times 1280 \approx 1.17 \times 10^9$ FLOPs.

- **Baseline (Total):** $3.89 \times 10^9 + 1.17 \times 10^9 \approx 5.06 \times 10^9$ FLOPs.

The resulting relative training overhead is:

$$\frac{\text{FLOPs}_{\text{NITP}}}{\text{FLOPs}_{\text{NTP}}} \approx \frac{1.18 \times 10^8}{5.06 \times 10^9} \approx 2.3\%. \tag{31}$$

*Table 6.* **Target-layer sweep across scales.** We report the average score over MMLU, MMLU-Pro, CSQA, BBH, and LCBench. The best target layer consistently lies in a shallow contextualized region around 20% of the total depth.

| Model / Setting | Target layer | Avg. |
|---|---|---|
| **3B MoE (17 layers)** | | |
| NTP baseline | – | 21.10 |
| Embedding target | 0 | 20.72 |
| Candidate target | 2 | 22.01 |
| **Selected target** | **4** | **23.58** |
| Candidate target | 6 | 22.12 |
| Candidate target | 8 | 21.22 |
| **9B MoE (24 layers)** | | |
| NTP baseline | – | 27.95 |
| Candidate target | 4 | 30.05 |
| **Selected target** | **5** | **30.75** |
| Candidate target | 6 | 29.80 |

## F.4. Empirical Wall-Clock Overhead

We also measure the practical training-time overhead in the same 9B MoE setting. Using the same number of GPUs and a global batch size of 1024, we compare NTP and NITP over a 5k-step run. The NTP baseline takes 16h 1m, while NITP takes 16h 18m, corresponding to an additional 17 minutes, or approximately 1.8% wall-clock overhead.

This empirical overhead is consistent with the FLOPs estimate above. The measured wall-clock increase is slightly smaller than the theoretical FLOPs ratio because large-scale MoE training also involves communication and system overheads that are largely unaffected by the NITP projection head and cosine loss.

**Conclusion.**  The analysis confirms that the overhead introduced by NITP remains marginal ($\sim 2\%$), ensuring scalability.

## G. Inference FLOPs Analysis

**Crucially, NITP introduces zero additional computational overhead during inference.** Since the NITP objective serves as an auxiliary supervision signal, the projection head is discarded after pre-training. The model architecture used for deployment remains identical to the standard transformer backbone. Therefore, the inference FLOPs per token are exactly the same as the baseline NTP model, ensuring that NITP improves model performance without compromising generation speed or increasing serving costs.

## H. Additional Ablation Studies

**Target-layer sweep.** We further study the choice of the layer used to construct the implicit target. As shown in Table 6, the best target consistently appears in a shallow but contextualized region: layer 4 out of 17 for the 3B MoE model and layer 5 out of 24 for the 9B MoE model, both close to 20% of the total depth. Using the embedding layer as the target is weaker, suggesting that purely lexical embeddings lack sufficient contextual information. Moving the target deeper also reduces the gain, likely because deeper representations become increasingly specialized for token discrimination and provide less complementary semantic supervision.

**Stop-gradient on implicit targets.** We ablate whether gradients are allowed to propagate into the implicit target representations in Equation (2). Allowing gradients to flow into the targets leads to co-adaptation between the predicted and target representations, which frequently results in unstable optimization and degraded performance. In contrast, applying a stop-gradient operation consistently stabilizes training and yields substantial improvements across downstream benchmarks. The results in Table 7 highlight that treating the implicit target as a fixed semantic anchor is critical for effective latent-space prediction, whereas disabling stop-gradient performs significantly worse, undermining the supervisory signal by allowing the target to drift during training. Accordingly, we adopt stop-gradient on implicit targets in all experiments.

**Start step of NITP.** Motivated by the three-phase dynamics observed in the NITP loss curve, we study when NITP

*Table 7.* **Additional ablations for NITP design choices.** We report results on key reasoning benchmarks. **Avg.** denotes the arithmetic mean of MMLU, MMLU-Pro, CSQA, BBH, and LCBench.

| Ablation Factor | Setting | MMLU | MMLU-Pro | CSQA | BBH | LCBench | Avg. |
|---|---|---|---|---|---|---|---|
| Baseline: NTP | | 34.60 | 11.00 | 34.15 | 21.92 | 3.82 | 21.10 |
| Stop Gradient | Disabled | 31.89 | 8.43 | 27.51 | 21.25 | 2.09 | 18.23 |
| | **Enabled** | **37.37** | **12.29** | **37.92** | **26.14** | **4.17** | **23.58** |
| NITP Start Step | 3000 | 35.54 | 10.14 | 31.85 | 21.25 | 2.09 | 20.17 |
| | 2000 | 36.17 | 10.28 | 34.72 | 23.03 | 2.78 | 21.40 |
| | 1000 | 35.93 | 11.57 | 31.22 | 23.40 | 4.00 | 21.22 |
| | **0** | **37.37** | **12.29** | **37.92** | **26.14** | **4.17** | **23.58** |
| Projector | × | 35.40 | 11.71 | 29.48 | 24.00 | 2.61 | 20.64 |
| | ✓ | **37.37** | **12.29** | **37.92** | **26.14** | **4.17** | **23.58** |

supervision should be activated during training. We compare enabling NITP from the beginning with introducing it at later stages. As shown in Table 7, delaying NITP activation leads to consistent performance degradation, particularly on reasoning-oriented benchmarks, while applying NITP from step 0 yields the best overall results. These results indicate that NITP is most effective when applied early, where it can shape representation geometry before higher-level semantics stabilize, thereby promoting more coherent alignment between shallow and deep representations throughout training.

**Effect of the projection head.** We study the role of the projection head in NITP by comparing our default design, which applies a two-layer MLP projector using SwiGLU on the last hidden states, with a projector-free variant that directly aligns the last hidden states with the implicit targets. Removing the projector consistently results in notable performance degradation across all benchmarks as shown in Table 7. We attribute this performance drop to a distributional mismatch between deep representations and shallow implicit targets. Directly enforcing alignment imposes an overly restrictive constraint, while a lightweight MLP projection head provides sufficient flexibility to bridge this mismatch without limiting the backbone capacity.

