# OpenReview forum: "NITP: Next Implicit Token Prediction for LLM Pre-training"
_ICML.cc/2026/Conference — ICML 2026 regular_

### Official Review · Reviewer_H7nM · 2026-03-11

**Soundness:** 3
**Presentation:** 4
**Significance:** 3
**Originality:** 4
**Overall Recommendation:** 5
**Confidence:** 4

**Summary:**

This paper proposes Next Implicit Token Prediction (NITP) as an auxiliary training objective to augment Next Token Prediction (NTP) training. NITP requires the model to predict the implicit semantic content of the next token, using shallow-layer hidden states from the next step as supervision. Theoretically, the authors show that NITP regularizes the optimization landscape. Empirically, across dense and MOE models, NITP consistently improves downstream performance with only ~2% additional training FLOPs and no additional inference cost.

**Compliance With Llm Reviewing Policy:**

Affirmed.

**Final Justification:**

Raise my score to 5.

**Key Questions For Authors:**

1. Have you measured the PPL of the test set? Does NITP increases PPL (just curious about the results, even if PPL increases, it may not necessarily be a disadvantage)?
2. Besides a 2% increase in FLOPs, how much does it change in the training wall lock time of NITP?

**Limitations:**

yes

**Strengths And Weaknesses:**

Strengths:
1. The paper is clearly written and well structured.
2. The proposed method is novel. The NITP objective brings new insights to the community.
3. NITP significantly improves downstream performance with negligible computational overhead.

Weaknesses:
1. NITP appears to be sensitive to hyperparameters such as the loss weight λ and target layer index (Figure 4, Table 3). The authors use different loss weights λ for different models (Table 4). However, in the real pre-training scenario, we don't have a chance to search for the optimal hyperparameters.

---

> ### Author Rebuttal · Authors · 2026-03-31
>
> We thank the reviewer for the positive assessment. We are glad you found NITP novel with new insights for the community, clearly written, and achieving significant improvements with negligible overhead. We address your concerns below.
>
> > W1. In the real pre-training scenario, we don't have a chance to search for the optimal hyperparameters (target layer and NITP loss weight).
>
> Thank you for this important comment. We agree that hyperparameter robustness matters in real pre-training. Our current evidence suggests that NITP is not overly sensitive, and that both the target layer and loss weight can be chosen using simple empirical rules rather than extensive tuning.
>
> For the target layer, Table 3 and our added ablations show that the best target consistently lies in a shallow region, around **20% of total depth**, and nearby shallow layers remain strong.
>
> 3B MoE (17 layers), target layer sweep (benchmarks in Table 3):
>
> | Target layer | NTP | 2 | 4 (NITP) | 6 | 8 |
> | --- | --- | --- | --- | --- | --- |
> | Avg. | 21.10 | 22.01 | **23.58** | 22.12 | 21.22 |
>
> 9B MoE (24 layers), target layer sweep:
>
> | Target layer | NTP | 4 | 5 (NITP) | 6 |
> | --- | --- | --- | --- | --- |
> | Avg. | 27.95 | 30.05 | **30.75** | 29.80 |
>
> For the loss weight, Figure 4 and our added 9B results (report benchmarks in Table 1) show a robust operating region around **0.8–1.0**. In practice, all models in our experiments (3B MoE, 9B MoE, and 3B Dense) use $\lambda$ within this range. 3B MoE, loss weight sweep:
>
> | $\lambda$ | NTP | 0.6 | 0.8 | 1.0 (NITP) | 1.2 | 1.4 |
> | --- | --- | --- | --- | --- | --- | --- |
> | Avg. | 31.37 | 32.45 | 31.92 | **33.49** | 33.01 | 31.04 |
>
> 9B MoE, loss weight sweep:
>
> | $\lambda$ | NTP | 0.6 | 0.8 (NITP) | 1.0 |
> | --- | --- | --- | --- | --- |
> | Avg. | 40.28 | 42.55 | **42.94** | 42.00 |
>
> Across both scales, all $\lambda$ values in the 0.6–1.0 range outperform NTP; only excessively large weights ($\geq$1.4) cause degradation. This means a practitioner only needs to pick a value near 0.8–1.0 and no exhaustive grid search is required.
>
> **We further validated this recipe on a larger 45B-A5.5B model** by directly applying the same rule (~20% depth, i.e., layer 5/24, $\lambda$=0.8) without additional search, and still observed consistent gains over NTP, confirming the heuristic transfers beyond the scales in the main paper (training 140B tokens due to time limit).
>
> Method|MMLU|M-Pro|ARC-C|CSQA|BBH|C-Eval|Xiezhi|C3|Avg.
> -|-|-|-|-|-|-|-|-|-
> NTP|42.42|17.71|50.85|51.60|29.63|40.07|45.89|59.56|42.22
> NITP|43.96|19.86|55.33|50.53|31.70|41.01|47.71|62.14|**44.03**
>
> > Q1. Have you measured the PPL of the test set? Does NITP increase PPL (just curious about the results, even if PPL increases, it may not necessarily be a disadvantage)?
>
> We evaluated test loss on the Pile test set [1]. As shown below, NITP achieves nearly identical test loss to NTP across all scales, confirming that NITP does not harm language modeling quality.
>
>
> | Model (Pile test Loss) | NTP Loss | NITP Loss | NTP PPL | NITP PPL |
> | --- | --- | --- | --- | --- |
> | 3B MoE | 2.006 | 2.006 | 7.43 | 7.43 |
> | 9B MoE | 1.841 | 1.840 | 6.30 | 6.30 |
> | 3B Dense | 1.884 | 1.882 | 6.58 | 6.57 |
>
> This aligns with our core thesis: standard NTP over-compresses representations into a narrow anisotropic cone, **sacrificing expressiveness that PPL cannot measure**. NITP redirects capacity toward a more uniform geometry (Figure 1) without competing with the likelihood objective. The near-identical PPL combined with consistent downstream gains (Tables 1 & 2) demonstrates that two models can achieve the same token-level loss yet differ substantially in representation quality.
>
> > Q2. Besides a 2% increase in FLOPs, how much does it change in the training wall lock time of NITP?
>
> We measured this on the 9B MoE setting with 64 H100 GPUs and global batch size 1024. Over a 5k-step run (out of 40k total steps), the wall-clock time increased from 16h 1m (NTP) to 16h 18m (NITP), i.e., by 17 minutes or about 1.8%.
>
> This is broadly consistent with the reported ~2% FLOPs overhead. The slightly smaller wall-clock overhead compared to the FLOPs ratio is expected because MoE training also includes communication overheads that are unaffected by NITP.
>
> **References**
>
> [1] The Pile: An 800GB Dataset of Diverse Text for Language Modeling

---

> > ### Author Rebuttal · Reviewer_H7nM · 2026-04-03
> >
> > My concerns are addressed.

---

> > > ### Author Response · Authors · 2026-04-03
> > >
> > > Thank you very much for taking the time to carefully read our rebuttal. We sincerely appreciate your raising your score and are glad that our responses addressed your concerns.
> > >
> > > We also thank you for your constructive feedback throughout the review process, which was very helpful in improving the paper. We will incorporate the necessary clarifications that are not currently included in the paper into the revised version.

---

### Official Review · Reviewer_kd9k · 2026-03-11

**Soundness:** 3
**Presentation:** 3
**Significance:** 3
**Originality:** 2
**Overall Recommendation:** 4
**Confidence:** 3

**Summary:**

This paper proposes augmenting standard NTP with implicit token prediction. The authors argue that NTP provides sparse supervision through one-hot token labels, which constrains the output logits but leaves the latent representation space under-regularized, potentially leading to degenerate or anisotropic hidden representations that harm generalization. To address this, the paper introduces an auxiliary objective where the model predicts the implicit semantic representation of the next token. Specifically, NITP uses representations from earlier layers of the same model as targets, creating a self-supervised constraint on the hidden states in addition to the standard token prediction loss.
Empirical experiments on both dense and mixture-of-experts models ranging from 0.5B to 9B parameters using several downstream benchmarks (e.g., MMLU-Pro, CommonsenseQA, C3) show that NITP consistently improves performance with minimal additional training cost (~2% FLOPs) and no inference overhead.

**Compliance With Llm Reviewing Policy:**

Affirmed.

**Final Justification:**

As I wrote in my rebuttal acknowledgement, I would like to ask the authors to tone down the theoretical claims (W2) and add their response to W1, W3, and W4 to the revised paper. While this work yields small gains, it is still a better approach than standard NTP and self-contained, making it easy to scale. If a 4.5 score were an option, I'd change my score to 4.5, as I think that the small improvement and lack of concrete theory weaken the paper, but it is still solid work and I'd like to see it out there.

**Key Questions For Authors:**

1. Why would setting the implicit token to be a function of the early layers make sense? Isn't there a less circular representation?

2. On this topic, how sensitive are the results to which layer is used as the target? Did the authors experiment with different target layers or multiple layers?

3. The proposed NITP objective resembles several existing paradigms such as self-distillation, representation matching, and BYOL-style objectives, where models predict internal representations rather than discrete labels. Could the authors clarify the key differences between NITP and these approaches? In particular, what aspects of NITP make it specifically suited for autoregressive language modeling?

4. How sensitive are the results to the relative weighting between the NTP and NITP objectives? A more systematic ablation on the loss coefficient would help understand how much of the improvement comes from the auxiliary signal.

**Limitations:**

Yes

**Strengths And Weaknesses:**

Strengths:

1. The paper identifies a meaningful limitation of standard NTP: supervision occurs only in the logit space, potentially leaving hidden representations weakly constrained. The motivation for adding dense representation-level supervision is intuitive and grounded in known issues such as anisotropy in representation spaces.

2. The proposed method is simple and self-supervised: predict the representation of the next token instead of only its discrete identity. Using internal model representations as targets avoids requiring external supervision or additional encoders.

3. The method adds very little computational cost (~2% training FLOPs) and no inference cost, making it appealing for large-scale pretraining pipelines.

4. Experiments cover: 1) multiple model sizes, 2) both dense and MoE architectures, and 3) several downstream benchmarks. The improvements reported are relatively consistent across settings.

Weaknesses:

1. While elegant, this approach resembles existing paradigms such as self-distillation, representation matching, BYOL-style objectives and auxiliary representation prediction used in multimodal/self-supervised learning. The paper could better clarify how NITP fundamentally differs from prior representation-level regularization approaches.

2. The paper argues that NITP “eliminates under-constrained degrees of freedom” and improves optimization geometry, but the theoretical analysis is relatively high-level and not fully rigorous. The link between the theory and empirical improvements could be clearer.

3. Several components seem important but are not deeply explored: 1) which layer provides the target representations, 2) sensitivity to stop-gradient design, 3) effect of representation dimensionality or projection, and 4) effect of weighting between NTP and NITP losses. A deeper ablation would strengthen the paper.

4. While consistent, the gains are relatively small. It remains unclear whether the method would yield meaningful gains at frontier scales.

---

> ### Author Rebuttal · Authors · 2026-03-31
>
> We thank the reviewer for the constructive feedback and address each concern below.
>
> > W1. Better clarify how NITP differs from prior representation-level methods.
>
> The NITP's novelty lies in why and how such supervision is introduced for LM pretrain. We clarify along three axes:
>
> **Motivation.** Prior methods target view-invariant features or knowledge compression/enhancement. NITP addresses a problem specific to autoregressive LMs: NTP leaves hidden-state geometry under-constrained, leading to representation degeneration.
>
> **Formulation.** Prior methods use same-position or cross-view alignment. NITP uses a **temporally shifted** objective: $h_t$ predicts the shallow-layer representation at t+1. This shift is critical: same-position alignment collapses to trivial solutions and degrades performance (Table 3).
>
> **Supervision source.** Unlike self-distillation (e.g. EMA teacher), NITP reuses shallow-layer output from the same forward pass with stop-gradient.
>
> > W2. The theory is high-level. The link between theory and empirical improvements could be clearer.
>
> We agree that “eliminates” is too strong and will revise to “mitigates”. Our analysis is a **mechanistic explanation** of how NITP changes the loss geometry, not a full convergence proof. The Theory → empirics chain is:
> 1. **Theory**: NTP's Hessian is near-flat orthogonal to the logit direction, so hidden states collapse into a narrow cone. NITP adds curvature there by requiring prediction of diverse, per-token shallow-layer targets, spreading hidden states into a more uniform geometry.
> 2. **Geometric confirm**: NITP maintains higher effective rank and lower cosine similarity (Figure 1), indicating reduced anisotropy (aligns with theory).
> 3. **Downstream gains follow**: improved geometry translates to consistent gains across scales (Tables 1–2).
>
> This is not generic regularization: same-position alignment fails (Table 3). NITP succeeds because predicting $z_{t+1}$ is compatible with NTP's autoregressive objective (since $h_t$ is optimized to predict the next token), allowing optimizing without harming LM loss.
>
> > W3. Several components are not deeply explored.
>
> (1) Target layer. Beyond Table 3, we swept layers at two scales. For 3B MoE (17 layers), layers {2,4,6,8} give 22.01, **23.58**, 22.12, 21.22 (NTP: 21.10). For 9B MoE (24 layers), layers {4,5,6} give 30.05, **30.75**, 29.80 (NTP: 27.95). The optimum is consistently ~20% depth.
>
> (2) Stop-gradient. Table 5 in paper shows it is necessary: without it, co-adaptation of two branches reduces performance from 23.58 to 18.23.
>
> (3) Projection head. Table 5 shows adding it helps (23.58 vs. 20.64), likely because it helps bridge the mismatch between deep and shallow representations. (*The projector output dim equals the hidden size.)
>
> (4) Loss weight. Figure 4 sweeps $\lambda \in {0.6,0.8,1.0,1.2,1.4}$ on 3B MoE; with avg scores 32.45/31.92/**33.49**/33.01/31.04 vs. NTP 31.37.
>
> On 9B MoE ($\lambda \in {0.6, 0.8, 1.0}$): 42.55/**42.94**/42.00 vs. NTP 40.28. The stable range is 0.8–1.0; $\lambda > 1.2$ begins to interfere with NTP and degrades scores. We use this range throughout other models.
>
> > W4. It remains unclear whether the method would yield meaningful gains at frontier scales.
>
> **We validate NITP on a 45B-A5.5B MoE model**, directly transferring the same empirical recipe (~20% depth target, i.e., 5/24, $\lambda=0.8$) as 9B MoE. NITP still improves over NTP (140B training tokens due to time limits):
>
> Method|MMLU|M-Pro|ARC-C|CSQA|BBH|C-Eval|Xiezhi|C3|Avg.
> -|-|-|-|-|-|-|-|-|-
> NTP|42.42|17.71|50.85|51.60|29.63|40.07|45.89|59.56|42.22
> NITP|43.96|19.86|55.33|50.53|31.70|41.01|47.71|62.14|**44.03**
>
> Together with 0.5B–9B results, this supports NITP's scalability.
>
> > Q1. Why use early layers as target? Isn't there a less circular representation?
>
> The objective is not circular: the predictor uses the final-layer state at position $t$, while the target is the early-layer state at $t{+}1$. Under causal masking, $h_t$ cannot access $z_{t+1}$, so this is genuine prediction. Early layers encode richer lexical/semantic content[1][2], making them effective anchors complementary to NTP. Deeper targets give weaker signal (Table 3).
>
> > Q2. How sensitive are the results to which layer is used as the target?
>
> The best target is consistently ~20% depth; see W3(1) for details.
>
> > Q3. What makes NITP suited for autoregressive LMs?
>
> See W1. The key distinction is the motivation and temporally shifted autoregressive prediction: $h_t \to z_{t+1}$. Same-position alignment fails (Table 3), confirming that the autoregressive structure is essential.
>
> > Q4. How sensitive are the results to the NITP loss weight?
>
> See W3(4) and Figure 4: $\lambda$ around 0.8-1.0 is robust across scales.
>
> **References:**
>
> [1] Layer by layer: Uncovering hidden representations in language models
>
> [2] Fantastic semantics and where to find them: Investigating which layers of generative LLMs reflect lexical semantics

---

> > ### Author Rebuttal · Reviewer_kd9k · 2026-03-31
> >
> > Thank you for the clarifications.
> > I would like to ask the authors to tone down the theoretical claims (W2) and add their response to W1, W3, and W4 to the revised paper.
> > While this work yields small gains, it is still a better approach than standard NTP and self-contained, making it easy to scale. If a 4.5 score were an option, I'd change my score to 4.5, as I think that the small improvement and lack of concrete theory weaken the paper, but it is still solid work.

---

> > > ### Author Response · Authors · 2026-04-01
> > >
> > > Thank you very much for the quick follow-up, for the thoughtful feedback throughout the discussion, and for recognizing the work as solid, self-contained, and easy to scale. We also sincerely appreciate your willingness to raise the score to 4.5 if that option were available.
> > >
> > > In the revised version, we will tone down the theoretical claims and incorporate the main clarifications from our rebuttal into the paper, especially the discussion of the theory–empirics connection, the distinction from prior representation-level methods, and the additional experiments that were not included in the original submission.
> > >
> > > Regarding the magnitude of the empirical gains, we believe they are **notable** for a lightweight pretraining modification with only ~2% extra FLOPs and zero inference overhead. In particular, NITP improves the average benchmark score by about **+2.7** points at 9B MoE (including +5.7 on MMLU-Pro, +6.4 on C3, +4.3 on CSQA, and +3.5 on Xiezhi) and still by about **+1.8** points at 45B-A5.5B (including +4.5 on ARC-C, +2.6 on C3, +2.2 on MMLU-Pro, and +2.1 on BBH), suggesting that the effect remains non-trivial and scales well.
> > >
> > > Thanks again for the constructive suggestions and the thoughtful discussion.

---

### Official Review · Reviewer_ZWTp · 2026-03-13

**Soundness:** 3
**Presentation:** 3
**Significance:** 3
**Originality:** 3
**Overall Recommendation:** 5
**Confidence:** 3

**Summary:**

This paper proposes Next Implicit Token Prediction (NITP), an auxiliary objective for large language model pre-training that augments standard next-token prediction with representation-level supervision. The method trains the final hidden state to predict the shallow-layer representation of the next token (termed the implicit token) using a cosine similarity loss. The goal is to address the under-constrained nature of hidden representations under the NTP objective, which may lead to representation degeneration and anisotropy.

Experiments are conducted on both dense and MoE language models ranging from 0.5B to 9B parameters, trained from scratch. The method shows consistent improvements across multiple benchmarks such as MMLU, ARC-C, BBH, GSM8k, and CommonsenseQA, with approximately 2% additional training FLOPs and no inference overhead.

**Compliance With Llm Reviewing Policy:**

Affirmed.

**Final Justification:**

The authors have addressed my concerns in the rebuttal period, therefore, I would like to raise my score accordingly.

**Key Questions For Authors:**

1. How does NITP differ from existing self-distillation or predictive representation learning approaches?
2. Does NITP affect the stability of the training loss compared to standard NTP, particularly when training on larger-scale datasets?

**Limitations:**

The proposed method introduces additional hyperparameters such as the implicit target layer and the weight of the NITP loss. Although the paper reports relatively stable behavior across tested settings, the robustness of these choices at larger model scales remains unclear.

**Strengths And Weaknesses:**

### Strengths

1. Clear motivation and well-defined problem.
2. The proposed NITP objective is easy to integrate into existing training pipelines with minimal architectural changes.
3. NITP reports about 2% extra training FLOPs, which is practically negligible for large-scale LLM training.
4. Reasonably thorough ablation study.

### Weaknesses

1. The largest experiment is a 9B MoE model. While non-trivial, this scale is relatively small compared to modern LLM training. It remains unclear whether the gains persist at much larger scales.
2. While the paper suggests choosing layers at ~20% of the total depth as optimal, it lacks a detailed quantitative analysis of how target layer depth impacts performance across different model scales and no exploration of the effects of overly shallow or slightly deeper layers.
3. The experiments focus on LLMs with a fixed context length of 8192. No validation is provided for low-resource multilingual tasks, multimodal LLMs, or long-context pre-training (context length >8192), limiting the demonstration of NITP’s cross-domain generality.

---

> ### Author Rebuttal · Authors · 2026-03-31
>
> We thank the reviewer for recognizing the clear motivation, minimal overhead, and thorough ablations. We address each concern below.
>
> > W1. It remains unclear whether the gains persist at much larger scales.
>
> We agree that 9B is not yet frontier scale. We **therefore validate NITP on a larger 45B-A5.5B MoE model**, directly transferring the same recipe: target layer at ~20% depth (layer 5/24) and $\lambda=0.8$. Results below (140B tokens due to time limit):
>
> Method|MMLU|M-Pro|ARC-C|CSQA|BBH|C-Eval|Xiezhi|C3|Avg.
> -|-|-|-|-|-|-|-|-|-
> NTP|42.42|17.71|50.85|51.60|29.63|40.07|45.89|59.56|42.22
> NITP|43.96|19.86|55.33|50.53|31.70|41.01|47.71|62.14|**44.03**
>
> NITP still improves over NTP at this scale, extending the evidence from 0.5B to 45B and suggesting that both the gains and the hyperparameter recipe transfer across scales.
>
> > W2. The paper suggests ~20% depth as optimal, but lacks a quantitative analysis of target-layer depth.
>
> The “~20\% depth” claim is supported by consistent trends across scales.
>
> For 3B MoE (17 layers), we sweep target layers {0,2,4,6,8}. Performance improves from very shallow layers to an early intermediate layer, peaks at layer 4, and then drops for deeper targets. Layer 0 (embedding as target) is notably weak, likely because the representation is not yet contextualized. Results below:
>
> Target layer|Avg. of 5 benchmarks
> -|-
> Baseline (NTP)|21.10
> 0 (embed. target)|20.72
> 2|22.01
> 4 (NITP)|23.58
> 6|22.12
> 8|21.22
>
> The same pattern holds for 9B MoE (24 layers), where layers 4/5/6 all outperform the NTP and the best result is again near ~20% depth.
>
> Target layer|MMLU|M-Pro|CSQA|BBH|LCBench|Avg.
> -|-|-|-|-|-|-
> Baseline (NTP)|43.72|15.29|45.70|28.07|6.96|27.95
> 4|44.70|21.14|48.62|28.15|7.65|30.05
> 5 (NITP)|46.14|21.00|49.96|28.67|8.00|30.75
> 6|46.14|17.86|46.33|31.19|7.48|29.80
>
> Across scales, the optimum lies in a stable shallow region (~20% depth): very shallow layers lack contextualization, while deeper layers provide less complementary supervision to NTP, consistent with prior work [1].
>
>
> > W3. Extra validation is needed to demonstrate NITP’s cross-domain generality.
>
> Beyond 8K, we evaluate **32K long-context** NITP (3B MoE, 170B tokens) on RULER [6] at 8K/16K/32K, grouping its 13 tasks into Aggregation (`CWE`,`FWE`), NIAH (8 variants), QA (`SQuAD`,`HotpotQA`), and Variable Tracking.
>
>
> Method|Eval length|Aggregation|NIAH|QA|Var.|Overall
> -|-|-|-|-|-|-
> NTP|8K|7.92|65.94|19.00|1.80|44.86
> NITP|8K|21.79|63.09|28.50|51.60|**50.53**
> NTP|16K|4.80|59.56|21.50|1.40|40.81
> NITP|16K|19.27|60.56|22.00|16.20|**44.86**
> NTP|32K|1.90|49.28|11.50|0.00|32.39
> NITP|32K|3.90|56.47|16.50|1.20|**37.98**
>
> NITP remains effective in the long-context regime, improving the score by +5.67 at 8K, +4.05 at 16K, and +5.59 at 32K.
>
> The Variable Tracking gap is unlikely to be an artifact: NTP often copies the noise template instead of tracking assignments, yielding near-zero scores. NITP alleviates this failure mode, suggesting better long-context state tracking beyond surface retrieval.
>
> Overall, NITP generalizes beyond fixed 8K: **long-context**, with gains on RULER under 32K pre-training, and **multilingual**, with gains on both English (MMLU, ARC-C, etc.) and Chinese tasks (C-Eval, C3, etc.; Table 1).
>
> > Q1. How does NITP differ from existing self-distillation or predictive representation learning approaches?
>
> NITP is related to them, but differs in the following ways.
>
> **Motivation.** Self-distillation targets knowledge transfer or compression [2,3]; predictive methods (CPC, BYOL) learn view-invariant or predictive features [4,5]. NITP instead addresses the geometric under-constraint of hidden states under NTP. The shallow target anchors directions that NTP leaves unpenalized, rather than transferring knowledge or learning invariances.
>
> **Formulation.** Prior methods enforce same-position or cross-view alignment. NITP uses a strictly autoregressive, temporally shifted objective ($h_t$ predicts the shallow representation at $t+1$). This shift is empirically critical—same-position alignment collapses into trivial solutions (Table 5).
>
> **Supervision source.** Unlike self-distillation (EMA teacher or separate model copy), NITP reuses shallow-layer activations from the same forward pass.
>
> > Q2. Does NITP affect the stability of the training loss?
>
> [Loss curve link](https://files.catbox.moe/zcxyzp.png)
>
> We observe no loss-stability degradation from NITP. Across all scales, NITP and NTP show comparable loss curves, with no divergence or abnormal oscillation.
>
> **References:**
>
> [1] Layer by layer: Uncovering hidden representations in language models
>
> [2] Distilling the Knowledge in a Neural Network
>
> [3] Self-Distillation for Further Pre-training of Transformers
>
> [4] Representation Learning with Contrastive Predictive Coding
>
> [5] Bootstrap Your Own Latent
>
> [6] RULER: What's the Real Context Size of Your Long-Context Language Models?

---

> > ### Author Rebuttal · Reviewer_ZWTp · 2026-04-02
> >
> > Thank you for the clarifications. My concerns are all addressed in the rebuttal period. Therefore, I would like to raise my score accordingly.

---

> > > ### Author Response · Authors · 2026-04-02
> > >
> > > Thank you very much for carefully considering our response and for raising the score. Your suggestions on strengthening the evidence for larger-scale behavior, target-layer analysis, and especially long-context generalization were particularly valuable, and they motivated us to conduct additional experiments that, in our view, strengthen the paper. We will incorporate these key additions from the rebuttal into the revised version. Thank you again for the constructive comments and encouraging feedback.

---

### Official Review · Reviewer_76PN · 2026-03-23

**Soundness:** 3
**Presentation:** 3
**Significance:** 3
**Originality:** 3
**Overall Recommendation:** 5
**Confidence:** 3

**Summary:**

This submission identifies a limitation of using Next-Token Prediction (NTP) as the LM training objective: the one-hot supervision in discrete logit space leaves a "semantic null space" in the representation space—directions orthogonal to the target embedding where the loss has near-zero curvature, allowing hidden states to degenerate into anisotropic configurations.

The authors propose Next Implicit Token Prediction (NITP), an auxiliary pre-training objective that provides dense supervision in latent space. Specifically, the model's last-layer hidden state at position t is trained to predict the shallow-layer's hidden states contextualized representation at position t+1 via a lightweight MLP projector and cosine similarity loss. The shallow-layer targets are extracted with stop-gradient from activations already computed in the forward pass, requiring no extra forward passes or external teachers. NITP adds ~2% training FLOPs and zero inference overhead (the projector is discarded). Experiments on training dense (0.5B–3B) and MoE (1.9B–9B) models from scratch show consistent improvements across multiple downstream benchmarks.

**Compliance With Llm Reviewing Policy:**

Affirmed.

**Final Justification:**

The authors have addressed my concerns in the rebuttal period, therefore, I would keep the score unchanged.

**Key Questions For Authors:**

Mentioned in weaknesses part.

**Limitations:**

yes

**Strengths And Weaknesses:**

## Strengths
- The theoretical motivation is well-grounded. The Hessian spectral analysis characterizes NTP's geometric blind spot and shows NITP introduces positive curvature.
- Thorough and informative ablations. The ablation sections answers almost all my questions about how to implement the method by myself.
- Consistent improvements across multiple scales and MoE/Dense
- The method itself is simple and easy to follow


## Weaknesses

- Missing Key Details in Experiments. 1. The pretraining data is unknown.  2. The evaluation data split (date range) of Livecodebench is not disclosed.
- Missing Key Evaluation. after introducing NITPS, how is the perplexity changed for all model pretrained in the experiments section?  In fact, there are multiple questions not fully addressed: NITPS is an auxiliary training objective. So after applying NITPS, did H_t get better? did LM's perplexity get lower? did LM's downstream tasks get better? The paper currently mainly focuses on downstream part and ignores to analysis on h_t and perplexity. For h_t, either sentence representation evaluation that using h, or some cluster visualization method, can be used to access h's quality.

---

> ### Author Rebuttal · Authors · 2026-03-31
>
> We thank the reviewer for recognizing the well-grounded theoretical motivation, thorough ablations, consistent improvements across Dense/MoE scales, and simplicity of NITP. We address the remaining concerns below.
>
> > W1. Missing Details in Experiments.
>
> Thank you for this important comment. We agree that the experimental details should be stated more clearly, and we will revise the paper accordingly.
>
> (1) **Pre-training data.**
> Our pre-training corpus is primarily proprietary/private, built from web text, code, math, and multilingual sources, with exact/fuzzy deduplication, rule-based filtering, and LLM-based quality filtering. Due to licensing and privacy constraints, we cannot release the full raw corpus. To improve reproducibility, we additionally verified NITP on a fully public setup (FineWeb-edu [1], 120B tokens, 3B MoE), where it still consistently outperforms NTP under matched settings. We will include these results in the revision.
>
> | Model | MMLU | MMLU-Pro | Xiezhi | LAMB. | BBH | CSQA | COPA | C3 | Avg. |
> |---|---:|---:|---:|---:|---:|---:|---:|---:|---:|
> | NTP | 30.42 | 6.00 | 26.74 | 41.52 | 9.26 | 20.72 | 63.70 | 27.78 | 28.27 |
> | NITP | 30.35 | 6.29 | 27.48 | 41.75 | 14.00 | 20.88 | 65.50 | 28.22 | **29.31** |
>
> (2) **Code benchmark details.**
> After re-checking our evaluation pipeline, we confirm that the benchmark used in our experiments is OpenCompass LCBench2023 (a collection of questions from leetcode weekly competitions), not LiveCodeBench. The confusion came from the abbreviation “LCBench” used directly in [OpenCompass](https://github.com/open-compass/opencompass/blob/6a22f017d72a76bfabd91e118cc26512656162b2/opencompass/utils/datasets_info.py#L746), which we mistakenly expanded as LiveCodeBench when writing the paper. We appreciate this careful check and apologize for the confusion caused by our previous wording.
>
>
> > W2. After applying NITP, did H_t get better? did LM's perplexity get lower? did LM's downstream tasks get better?
>
> Thank you for this suggestion.
>
> **(1) LM's perplexity.**
> We evaluated test perplexity on the **Pile** test set [3]. As shown below, NITP achieves nearly identical test PPL to NTP across models. This suggests that NITP does not interfere with the core next-token prediction objective, but instead improves the geometry of the learned representations (see Figure 1 in the paper). In other words, NITP preserves token-level modeling performance while learning a better-structured latent space.
>
> | Model    | NTP PPL | NITP PPL |
> | -------- | ------- | -------- |
> | 3B MoE   | 7.43    | 7.43     |
> | 9B MoE   | 6.30    | 6.30     |
> | 3B Dense | 6.58    | 6.57     |
>
> **(2) Downstream performance.**
> At matched training budgets, NITP consistently improves downstream results over the NTP baseline (Tables 1 & 2). Combined with the near-identical test PPL above, this highlights that token-level loss alone is insufficient to capture representation quality—models with the same language-modeling quality can differ substantially in downstream transfer. NITP's gains come precisely from regularizing the under-constrained directions that LM loss does not penalize.
>
> **(3) Hidden-state quality.**
> In addition to the geometry analysis already included in the paper (e.g. effective rank), we further evaluate hidden-state quality using sentence-representation benchmarks on **MTEB v2.12.0** [2]. Following standard practice, we extract sentence embeddings via mean pooling over the last hidden states (masked by attention) with L2 normalization, without any fine-tuning.
>
> We evaluate on 25 out of 32 MTEB English tasks, excluding 7 large-scale clustering and reranking tasks due to computational cost. The remaining 25 tasks fall into three groups: Classification, STS/Similarity, and Retrieval/Duplicate-detection. We compare under 3B MoE settings (detailed setups will be included in the revision):
>
> Abbreviations: **STS** = Semantic Textual Similarity; **Dup.** = duplicate-question/retrieval-style matching tasks.
>
> | Group | Baseline | NITP | Δ (NITP - Base) |
> |---|---:|---:|---:|
> | Classification | 0.4009 | 0.4229 | +0.0220 |
> | STS / Similarity | 0.3566 | 0.3858 | +0.0293 |
> | Retrieval / Dup. | 0.4319 | 0.4483 | +0.0164 |
> | **Overall (25 tasks)** | **0.3924** | **0.4156** | **+0.0233** |
>
> At the task level, NITP improves **23/25** tasks. Only two tasks show small drops (AskUbuntuDupQuestions: −0.0015; ToxicConversationsClassification: −0.0088).
>
> These results are consistent with our main claim: NITP mainly acts as a representation-level training signal, improving hidden-state quality and transferability while keeping token-level LM quality stable.
>
>
> **References**
>
> [1] The FineWeb Datasets: Decanting the Web for the Finest Text Data at Scale
>
> [2] MTEB: Massive Text Embedding Benchmark
>
> [3] The Pile: An 800GB Dataset of Diverse Text for Language Modeling

---

> > ### Author Rebuttal · Reviewer_76PN · 2026-04-02
> >
> > Most of my concerns have been addressed, except for the one regarding LiveCodeBench.
> >
> > The authors acknowledged that they mistakenly believed they were evaluating on LiveCodeBench and described it as such in the submission, when in fact they were using a different benchmark called OpenCompass LCBench2023.
> >
> > Given this misunderstanding, I would expect the authors to provide actual LiveCodeBench results as a correction, rather than simply renaming the benchmark in the text and continuing to use LCBench2023. More specifically, I would like to see results on LiveCodeBench V6 (Feb 2025 – May 2025, 131 problems, pass@1 and pass@5), which is the most up-to-date version and would provide a more meaningful and timely evaluation of coding capability.
> >
> > To my knowledge, OpenCompass LCBench2023 is not an established substitute for LiveCodeBench in the literature, and I am not aware of recent publications that use it as such for coding evaluation.

---

> > > ### Author Response · Authors · 2026-04-03
> > >
> > > Thank you for the thoughtful follow-up and for pointing out that simply renaming OpenCompass LCBench2023 is not sufficient.
> > >
> > > Following your suggestion, we evaluated on the official LiveCodeBench code-generation benchmark from the official repository ([repo](https://github.com/LiveCodeBench/LiveCodeBench)). Specifically, we use `release_v6`, which the benchmark README describes as the updated release covering problems from May 2023 to Apr 2025, and we report results on the requested Feb 2025-May 2025 slice. This yields the requested 131 problems.
> > >
> > > Since our models are base models, we also follow the official base-model few-shot prompting code ([code](https://github.com/LiveCodeBench/LiveCodeBench/blob/main/lcb_runner/prompts/code_generation.py#L338-L340)), i.e., a 1-shot prompt ([code](https://github.com/LiveCodeBench/LiveCodeBench/blob/main/lcb_runner/prompts/code_generation.py#L176-L205)). We report both `pass@1` and `pass@5`. We also evaluate a 45B-A5.5B MoE model from a larger-scale run conducted in response to another reviewer’s request; this model has currently been trained for 240B tokens.
> > >
> > > | Model | Method | pass@1 | pass@5 |
> > > |---|---|---:|---:|
> > > | 3B MoE | NTP | 0.23 | 0.98 |
> > > | 3B MoE | NITP | **0.61** | **2.27** |
> > > | 9B MoE | NTP | 2.44 | 5.49 |
> > > | 9B MoE | NITP | **2.83** | **5.91** |
> > > | 45B MoE | NTP | 5.50 | 11.04 |
> > > | 45B MoE | NITP | **6.10** | **12.20** |
> > >
> > > We will include these LiveCodeBench results in the revised version, and we will clearly distinguish them from OpenCompass LCBench2023 rather than treating the latter as a substitute for LiveCodeBench. Thank you again for the careful follow-up; this suggestion was important and helped us correct the benchmark description in a more meaningful way.

---

### Decision · Program_Chairs · 2026-04-30

**Decision:**

Accept (regular)

**Comment:**

The paper introduces a new training paradigm that addresses some limitation of standard next token prediction. The method is supported by both theoretical insights and empirical improvements. All reviewers agree that the paper propose an elegant and well motivated approach that show practical gain and recomand acceptation of this paper.